# ExGRPO: Learning to Reason from Experience

**Runzhe Zhan**[12*]  **Yafu Li**[24✉]  **Zhi Wang**[32]  **Xiaoye Qu**[2]  **Dongrui Liu**[2]  **Jing Shao**[2]
**Derek F. Wong**[1✉]  **Yu Cheng**[4]
[1]University of Macau   [2]Shanghai AI Laboratory   [3]Nanjing University
[4]The Chinese University of Hong Kong
nlp2ct.runzhe@gmail.com   yafuly@gmail.com   zhiwang@nju.edu.cn
{quxiaoye, liudongrui, shaojing}@pjlab.org.cn
derekfw@um.edu.mo   chengyu@cse.cuhk.edu.hk

## Abstract

Reinforcement learning from verifiable rewards (RLVR) is an emerging paradigm for improving the reasoning ability of large language models. However, standard on-policy training discards rollout experiences after a single update, leading to computational inefficiency and instability. While prior work on RL has highlighted the benefits of reusing past experience, the role of experience characteristics in shaping learning dynamics of large reasoning models remains underexplored. In this paper, we are the first to investigate what makes a reasoning experience valuable and identify rollout correctness and entropy as effective indicators of experience value. Based on these insights, we propose **ExGRPO** (**Ex**periential **G**roup **R**elative **P**olicy **O**ptimization), a framework that organizes and prioritizes valuable experiences, and employs a mixed-policy objective to balance exploration with experience exploitation. Experiments on five backbone models (1.5B-8B parameters) show that ExGRPO consistently improves reasoning performance on mathematical/general benchmarks, with an average gain of +3.5/7.6 points over on-policy RLVR. Moreover, ExGRPO stabilizes training on both stronger and weaker models where on-policy methods fail. These results highlight principled experience management as a key ingredient for efficient and scalable RLVR.

Code: 🎮 ExGRPO        Models: 🤗 ExGRPO Models

## 1 Introduction

Reinforcement learning (RL) has become a pivotal technique for advancing the reasoning capabilities of language models on complex tasks (Guo et al., 2025; Yang et al., 2025). RL augments language models' reasoning by modeling chain-of-thought (CoT) as action sequences optimized under verifiable rewards (RLVR), laying the groundwork for sophisticated downstream applications (Gridach et al., 2025). However, a significant yet overlooked challenge persists: due to the on-policy nature of most RLVR algorithms, the valuable experience generated during the rollout phase is often discarded after a single gradient update (Shao et al., 2024). This practice not only squanders substantial computational resources but also forfeits critical opportunities for the model to learn from prior successful explorations, thereby imposing a bottleneck on scaling RL for reasoning.

Experience replay (Lin, 1992) is a widely adopted technique in RL to address this issue and improve sample efficiency. As stated in Silver & Sutton (2025), AI is at the cusp of a new period in which experience will become the dominant medium of improvement. This idea, often referred to as experience-based RL, leverages the intuition that previously collected interactions (i.e., state-action-reward tuples) contain valuable information for learning, helping models stabilize training and accelerate convergence (Mnih et al., 2013). A non-trivial challenge lies in how to exploit past experiences based on their differing "values" (Schaul et al., 2016) and manage the replay process according to customized learning schedules (Sujit et al., 2023). However, efficient experience replay mechanisms remain largely underexplored in RLVR for building large reasoning models (LRMs;

---

* Work was done during Runzhe Zhan's internship at Shanghai AI Laboratory.
✉ Corresponding authors.

Plaat et al. 2024). Given the vast quantity of experiences collected during rollouts, a fundamental question remains: *How can the reasoning model effectively exploit its own stream of experience to maximize learning toward scaling RL compute for LRMs?*

To address this gap, we begin by examining what constitutes a valuable reasoning experience for RLVR optimization[1]. We hypothesize that experience utility varies with measurable properties. RLVR experience generally consists of a question and its corresponding trajectories. Accordingly, we study experience properties from these two components. Through systematic analysis, we identify rollout correctness (for questions) and trajectory entropy (for trajectories) as effective online proxy metrics for characterizing experience quality. Specifically, tasks of intermediate difficulty and their associated low-entropy trajectories tend to be beneficial for RLVR optimization.

Building on these insights, we introduce **Ex**periential **G**roup **R**elative **P**olicy **O**ptimization (ExGRPO), a novel framework designed to strategically identify, manage, and replay valuable experiences. ExGRPO maintains a replay buffer of reasoning trajectories derived from partially correct rollouts and organizes them into buckets according to their correctness levels. To manage the buffer effectively, it uses a sampling strategy that prioritizes experiences from the most beneficial buckets, along with the corresponding trajectory with the lowest entropy. This approach allows the model to learn more efficiently from past experiences that align best with its current capabilities, guided by the principles of valuable experience concluded in our preliminary analysis. During mini-batch optimization, ExGRPO uses a mixed-policy optimization objective that balances between leveraging fresh exploration and reusing strategically selected past experiences, improving both sample efficiency and training stability.

Our experimental results demonstrate that ExGRPO delivers improvements over on-policy RLVR baselines. We evaluate across five backbone models, spanning the Qwen (Yang et al., 2024; Qwen et al., 2024) and Llama (Grattafiori et al., 2024) families from 1.5B to 8B parameters, on both mathematical reasoning benchmarks (AIME24/25, AMC, OlympiadBench, Minerva, MATH500) and out-of-distribution reasoning benchmarks (ARC-c, GPQA, MMLU-Pro). Averaged over **all backbone models**, ExGRPO achieves +3.5 and +7.6 point gains[2] on in-distribution and out-of-distribution benchmark performance, respectively, compared to the on-policy RLVR. Notably, ExGRPO stabilizes RLVR training on the weaker Llama-3.1 8B model (Grattafiori et al., 2024) and continual learning on the stronger LUFFY model (Yan et al., 2025), where on-policy optimization collapses. Detailed analysis and ablations further confirm that improvements stem from ExGRPO's experience management and optimization mechanisms, which amplify the utility of past explorations.

## 2 RELATED WORK

**Reinforcement Learning with Verifiable Rewards.** Reinforcement Learning with Verifiable Rewards (RLVR; Lambert et al. 2024) frames the language model as a policy that generates reasoning trajectories for verifiable tasks, with rewards provided by rule-based (Guo et al., 2025; Yu et al., 2025) or model-based verifiers (Su et al., 2025; Ma et al., 2025b; Chen et al., 2025a). Most RLVR methods adopt on-policy optimization (Schulman et al., 2017; Shao et al., 2024), which ensures stability but incurs high computational cost. Recent work has explored off-policy techniques (Kallus & Uehara, 2020; Meng et al., 2023) that incorporate historical or external data. Some approaches mix expert demonstrations with on-policy updates, e.g., off-policy policy gradients (Yan et al., 2025), SFT loss (Zhang et al., 2025b; Ma et al., 2025a), or knowledge distillation (Xu et al., 2025); others develop direct off-policy update rules for improved sample efficiency (Cohen et al., 2025; Roux et al., 2025; Arnal et al., 2025). However, they overlook the impact of data quality within the experience replay buffer, a factor that remains underexplored in RLVR and is the central focus of our work.

**Experience-based Reinforcement Learning.** Experience replay (Lin, 1992) is a classical RL technique to improve sample efficiency and stabilize training, later extended with prioritized replay (Schaul et al., 2016; Sujit et al., 2023) to emphasize informative transitions, enabling breakthroughs in control tasks (Mnih et al., 2013; 2015; Lillicrap et al., 2016). For LRMs, recent studies show that replaying successful trajectories accelerates convergence and improves reasoning capabilities. ReMix (Liang et al., 2025) leverages off-policy experience replay to improve training efficiency; RePO adopts online replay (Li et al., 2025), collecting early on-policy rollouts and revisiting them asynchronously; RLEP (Zhang et al., 2025a) reuses trajectories from a well-trained policy; and

---

[1]In the context of RLVR, the experience refers to a state-action-reward trajectory during rollout of the reasoning chain. We will use the terms experience/trajectory/rollout interchangeably throughout the paper.

[2]Due to space limitations, *numerical results* of some backbone models are reported in Section E.3.

RRL (Dou et al., 2025) dynamically revisits promising early states; ARPO (Lu et al., 2025) extends this idea to GUI agents, combining GRPO with a replay buffer to enhance stability and sample efficiency. Among them, the last three methods overlook importance weighting to correct distribution mismatch in off-policy updates. In this work, we argue that not all stored experiences are equally valuable in zero RLVR for LRMs. We systematically analyze properties that determine the value of past experiences and design methods to better leverage high-value experiences, thereby improving efficiency and performance.

## 3 PRELIMINARIES

In this section, we briefly review the RL foundations underlying our method: RLVR and Group Relative Policy Optimization (GRPO), upon which ExGRPO is built. We then present a preliminary analysis of experience data, i.e., the model's past successful rollouts, and examine how their characteristics influence performance.

### 3.1 REINFORCEMENT LEARNING WITH VERIFIABLE REWARD

**Verifiable Reward Function.** The verifiable reward compares the extracted answer from the model's output with a predefined golden answer. For example, the model is instructed to output the final answer in a format such as \boxed{}, from which a verifier extracts the value. Formally, given a model output $o$ for question $q$, the reward is:

$$r(q, o) = \begin{cases} 1 & \text{if } o \text{ contains the correct final answer to } q \\ 0 & \text{otherwise.} \end{cases} \tag{1}$$

**Group Relative Policy Optimization (GRPO).** GRPO (Shao et al., 2024) is a strong baseline within the RLVR paradigm (Guo et al., 2025; Zeng et al., 2025; Liu et al., 2025), achieving effective scaling without requiring an additional value model. It estimates the advantage of each trajectory by normalizing rewards within a group of $N$ sampled solutions. Given a group of $K$ trajectories $\mathcal{G}_q = \{o_i\}_{i=1}^K$ using the current reference policy $\pi_{\theta_{\text{old}}}$ (i.e, rollout policy), GRPO estimates the advantage of each trajectory $o_i$ by normalizing its reward against the empirical statistics of the group:

$$\widehat{A}_i = \frac{r(q, o_i) - \mu_{\mathcal{G}_q}}{\sigma_{\mathcal{G}_q}} \tag{2}$$

where $\mu_{\mathcal{G}_q} = \frac{1}{K} \sum_{j=1}^K r(q, o_j)$ and $\sigma_{\mathcal{G}_q}$ are the mean and standard deviation of rewards in the group $\mathcal{G}_q$. The on-policy objective maximizes a clipped surrogate function:

$$\mathcal{J}_{\text{GRPO}}(\theta) = \mathbb{E}_{q \sim \mathcal{D}, \{o_i\} \sim \pi_{\theta_{\text{old}}}(\cdot|q)} \left[ \frac{1}{K} \sum_{i=1}^K \text{CLIP}(w_i(\theta), \widehat{A}_i) \right] \tag{3}$$

where $\mathcal{D}$ is the training query set and the loss term is $\text{CLIP}(w, A) = \frac{1}{|o|} \sum_{t=1}^{|o|} \min(w_t A, \text{clip}(w_t, 1 - \varepsilon, 1 + \varepsilon)A)$. Since the trajectory $o$ is generated by the rollout policy model before update, i.e., $\pi_{\theta_{\text{old}}}$, the per-token importance weight $w_{i,t}(\theta)$ is the probability ratio between the current policy $\pi_\theta$ and reference policy $\pi_{\theta_{\text{old}}}$: $w_{i,t}(\theta) = \pi_\theta(o_{i,t}|q, o_{i,<t})/\pi_{\theta_{\text{old}}}(o_{i,t}|q, o_{i,<t})$. Following Dr.GRPO (Liu et al., 2025), we remove the length normalization and standard deviation normalization of GRPO loss (Eqs. 2 and 3).

### 3.2 PRELIMINARY STUDY ON EXPERIENCE DATA

To motivate our experience management design, identifying which questions and trajectories are most valuable for experience-based learning, we begin by analyzing rollouts from on-policy RLVR. Our study is guided by two questions: (1) Are all questions equally useful for training, or do certain difficulty levels provide stronger learning signals? and (2) Does low entropy correlate with higher-quality trajectories that the model should prioritize?

**Setup.** We conduct experiments using Qwen2.5-Math 7B (Yang et al., 2024) backbone model, trained with vanilla on-policy RLVR following the Dr.GRPO setup on the OpenR1-Math dataset (Face, 2025). Implementation details and hyperparameters are provided in Section 5.1. The experimental setting is as follows: we train three models, each using training batches restricted to questions of a particular difficulty level, determined by rollout correctness of each question. During training, we collect the generated trajectories and analyze their characteristics, such as entropy and the quality (validity) of the underlying reasoning chains. The entropy refers to the average action entropy: $H(o) = -\frac{1}{|o|} \sum_t \pi(o_t \mid q, o_{<t}) \log \pi(o_t \mid q, o_{<t})$, indicating the uncertainty of the model generating each action (token). Under this setting, three models are trained with a comparable number of samples, and detailed statistics are provided in Appendix F.1.

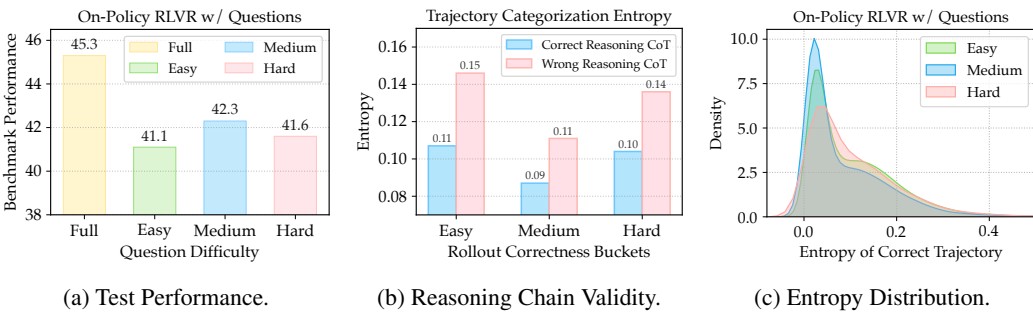

|               (a) Test Performance.                |               (b) Reasoning Chain Validity.                |               (c) Entropy Distribution.                |

Figure 1: Analysis of question difficulty and entropy in on-policy RLVR training: (a) Test performance of models trained on different question groups; (b) Entropy comparison between correct and incorrect trajectories; (c) Entropy distributions of logically correct trajectories across question groups;

**Are all questions equally useful for training?** During training, we categorize each question $q$ into one of three difficulty buckets based on its online (rollout) correctness rate: *Easy* $[75\%, 100\%]$, *Medium* $(25\%, 75\%]$, and *Hard* $(0, 25\%]$. This classification is computed online without additional decoding or offline annotation. We implement bucket-specific experiments by restricting each batch to questions from a single difficulty group, producing three models, along with one trained on the full set. Results in Figure 1a show that training on Medium questions yields the largest performance gains. In contrast, models trained exclusively on other questions perform substantially worse than the full-data baseline, yet they still provide complementary signals and should not be entirely discarded.

**Does lower entropy imply better reasoning?** We then test which metric can serve as an online proxy for reasoning quality. While outcome-based rewards check the final answer, they do not capture whether the reasoning CoT is logically correct or valid. To establish a reference, we follow previous work (Wen et al., 2025) to employ an LRM (Qwen3-32B; Yang et al. 2025) as an external judge, which inspects correct outputs and labels their reasoning validity (See Section F.2). As shown in Figure 1b, correct reasoning trajectories exhibit lower entropy than incorrect ones, indicating that selecting the lowest-entropy candidate generally yields higher CoT quality. We further examine the entropy distribution of correct trajectories across the three question groups in Figure 1c. Correct trajectories in the Medium group concentrate at lower entropy values, while Easy and Hard groups display broader distributions, with Hard problems peaking at higher entropy. This observation partly explains the marginal advantage of training on Medium-level questions compared to the other groups.

In summary, we highlight two guidelines: *Medium-difficulty **questions** provide the most valuable optimization signals*, and *entropy minimization is an effective heuristic for **trajectory** selection*.

**Low-entropy trajectories mitigate misguided learning.** Entropy magnitude (Chen et al., 2025b) is a key signal in RLVR training. Prior work shows that aggressively minimizing entropy can cause collapse (Cui et al., 2025b), motivating a more balanced exploration–exploitation strategy in experience management. While high-entropy policies encourage exploration, they can yield low-quality trajectories, many successes are merely lucky hits from incorrect reasoning chains.

Repeatedly sampling such misguided experiences from the replay buffer contaminates training, leading to systematic reasoning errors, we refer to a *snowball effect*. For instance, as illustrated in

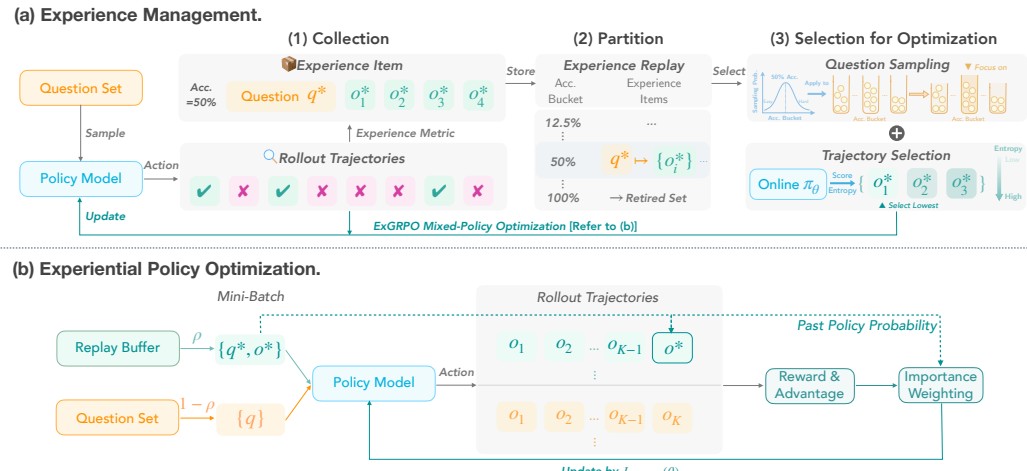

Figure 2: Overview of **Ex**periential **G**roup **R**elative **P**olicy **O**ptimization (ExGRPO). ExGRPO operates in two phases: (a) Experience Management and (b) Policy Optimization (cf. Algorithm 1).

Section F.4, we observe models using invalid code blocks for math problems, traced to high-entropy trajectories in the buffer. Thus, entropy minimization should be a guiding principle for experience selection, especially early in training when stability matters most. However, balancing entropy reduction with exploration remains a non-trivial challenge, which we will detail in the methodology.

## 4 METHODOLOGY

Based on previous findings, we propose Experiential Group Relative Policy Optimization (ExGRPO) as shown in Figure 2, a method that leverages structured experience replay with entropy-based trajectory selection to enhance sample efficiency while preserving policy stability.

### 4.1 ExGRPO: EXPERIENCE MANAGEMENT

**Problem Formulation.** Given a dataset $\mathcal{D}$ and a reward model $r(\cdot)$, ExGRPO iteratively updates a policy model $\pi_\theta$ by combining on-policy rollouts $\mathbb{E}_{q\sim D, o\sim \pi_{\theta_{old}}(\cdot|q)}\big[r(q, o)\big]$ and experience replay $\mathbb{E}_{q^*\sim \mathcal{E}, o^*\sim \pi_{\theta_{past}}(\cdot|q^*)}\big[r(q^*, o^*)\big]$. The $\pi_{\theta_{old}}$ is the rollout policy model before gradient updates, while $\pi_{\theta_{past}}$ is the previous one which generates the corresponding experiential trajectories. The buffer $\mathcal{E}$ is maintained as a key-value structure $q^* \mapsto \{o^*\}$, where each question is associated with a set of candidate trajectories. We next describe the three-stage experience management pipeline.

**Experience Collection.** During training, we collect the model's successful trajectories for each question in the batch. For a given question $q$ sampled from the dataset, the rollout policy $\pi_{\theta_{old}}$ generates $K$ trajectories, denoted as $\mathcal{G}_q^* = \{o_i^*\}_{i=1}^K$. We then compute the correctness rate $\text{Acc}(q^*) = k/K$, where $k$ is the number of successful trajectories verified by the reward model. All successful rollouts are stored in the replay buffer $\mathcal{E}$, recorded as a mapping $q^* \mapsto \{o^*\}$. The associated correctness rate is saved together with the query and later used for experience partition.

**Experience Partition.** The replay buffer $\mathcal{E}$ is partitioned into buckets by each question's **latest** correctness rate $\text{Acc}(q^*)$, with $q^*$ as the indexing key. Prior work (Zhang & Zuo, 2025) reweights advantages based on correctness rates, whereas we leverage them to bucket experiences by difficulty. To prevent overfitting on trivial cases, we introduce a *Retired Set*: questions solved in all rollouts are removed from $\mathcal{E}$, ensuring optimization focuses on partially solved questions. Xiong et al. (2025) also filters prompts by online correctness. In contrast, our retired-set strategy excludes mastered experience items.

**Experience Selection.** During optimization, ExGRPO retrieves a mini-batch of experiential samples from partitioned buffer $\mathcal{E}$ in two steps:

*(1) Question Sampling.* To prioritize experiences described in Section 3.2, each bucket is sampled with probability $p \propto \mathcal{N}(\text{Acc}(q^*);\ \mu,\ \sigma = 1)$, where $\mu$ is set to $0.5$ in practice. This ensures that medium-difficulty questions are sampled more often than trivially easy or mostly failed ones, allowing biases sampling towards the "sweet spot" questions.

*(2) Trajectory Selection.* For each sampled question, we select the trajectory with the lowest entropy **under the current policy** without relying on an external judge, i.e., $o^* \leftarrow \arg\min_{o_i \in \{o^*\}} H(o_i; \pi_\theta)$, thereby prioritizing trajectories which indicate better CoT quality as discussed in Section 3.2.

After three-stage *management*, selected experiences are used for mixed-policy *optimization*.

## 4.2 ExGRPO: Experiential Policy Optimization

The core idea of ExGRPO optimization is to unify on-policy exploration with off-policy replay under a joint objective, while correcting the distribution shift. At each step, a mini-batch $\mathcal{B}$ is constructed from: (1) *on-policy samples* $\mathcal{B}_{\text{on}}$ from dataset $\mathcal{D}$, and (2) *experiential samples* $\mathcal{B}_{\text{exp}}$ from buffer $\mathcal{E}$. A hyperparameter $\rho \in [0, 1)$ determines the experiential proportion, with $|\mathcal{B}_{\text{exp}}| = \min(\lfloor \rho B \rfloor, |\mathcal{E}|)$ and $|\mathcal{B}_{\text{on}}| = B - |\mathcal{B}_{\text{exp}}|$. This ensures a lower bound in cases where the experience replay buffer is depleted. The overall ExGRPO objective is a combination of two components:

The *on-policy objective* $\mathcal{J}_{\text{on}}(\theta)$ follows GRPO in Eq. 3: for each query $q \sim \mathcal{B}_{\text{on}}$, $K$ trajectories $\{o_i\}_{i=1}^K$ are sampled from policy $\pi_{\theta_{\text{old}}}$ to form an advantage group $\mathcal{G}_q$, with objective $\mathcal{J}_{\text{GRPO}}(q, \mathcal{G}_q; \theta)$.

For the *experiential off-policy objective*, given $q^* \sim \mathcal{B}_{\text{exp}}$, we form a mixed advantage estimation group $\mathcal{G}_{q^*} = \{o^*\} \cup \{o_i\}_{i=1}^{K-1}$, where $o^*$ is a replayed trajectory from past policy $\pi_{\theta_{\text{past}}}$ and $\{o_i\}$ are new rollouts from current reference policy $\pi_{\theta_{\text{old}}}$. To obtain an unbiased policy gradient estimate from the replayed trajectory, $o^*$ is reweighted by $w_t^*(\theta) = \frac{\pi_\theta(o_t^* | q^*, o_{<t}^*)}{\pi_{\theta_{\text{past}}}(o_t^* | q^*, o_{<t}^*)}$. All trajectories in $\mathcal{G}_{q^*}$ use GRPO advantage estimates $\widehat{A}$ in Eq. 2. Finally, the unified objective is:

$$
\mathcal{J}_{\text{ExGRPO}}(\theta) = \underbrace{(1 - \rho) \cdot \mathbb{E}_{q \sim \mathcal{B}_{\text{on}}} \left[ \frac{1}{K} \sum_{i=1}^K \text{CLIP}(w_i(\theta), \widehat{A}(o_i, \mathcal{G}_q)) \right]}_{\text{Expansion of On-policy objective } \mathcal{J}_{\text{GRPO}}(q, \mathcal{G}_q; \theta)}
$$

$$
+ \underbrace{\rho \cdot \mathbb{E}_{q^* \sim \mathcal{B}_{\text{exp}}} \left[ \frac{1}{K} \left( \text{CLIP}(w^*(\theta), \widehat{A}(o^*, \mathcal{G}_{q^*})) + \sum_{i=1}^{K-1} \text{CLIP}(w_i(\theta), \widehat{A}(o_i, \mathcal{G}_{q^*})) \right) \right]}_{\text{Experiential Off-Policy objective}}
$$

$$(4)$$

Directly optimizing on replayed trajectories, which are often selected for their low entropy, can hurt the exploration in the RLVR. We introduce two mechanisms to ensure stable and effective learning:

**Policy Shaping for Entropy Preservation.** Inspired by Yan et al. (2025), we modulate the gradient from experiential data using policy shaping. We replace "CLIP" term of importance sampling for $o^*$ in Eq. 4 with $f(w^*(\theta)) \cdot \widehat{A}(o^*, \mathcal{G}_{q^*})$, where $f(x) = \frac{x}{x+\beta}$ is a non-linear transformation with a small constant $\beta = 0.1$. This amplifies low-probability signals and dampens high-probability ones within the replayed trajectory, encouraging the model to learn from its more novel aspects from experience.

**Delayed Start Mechanism.** To mitigate the collection of low-quality trajectories during the initial training stages when the model's capabilities are still developing, we employ a delayed start. The on-policy RLVR process runs first, and ExGRPO is activated only after the *Pass@1* metric for a training batch surpasses a predefined threshold, ensuring experiential samples are of higher quality.

## 5 Experiments

### 5.1 Experimental Setup

**Data and Evaluation.** We train on the OpenR1-Math 45k subset (Face, 2025; Yan et al., 2025) and evaluate on nine challenging reasoning benchmarks. Six are in-distribution math benchmarks:

Table 1: Overall in-distribution and out-of-distribution performance based on Qwen2.5-Math-7B. **Bold** and underline indicate the best and second-best results within a comparable group, respectively.

| Model | In-Distribution Performance | | | | | | | Out-of-Distribution Performance | | | |
|---|---|---|---|---|---|---|---|---|---|---|---|
| | AIME24 | AIME25 | AMC | MATH-500 | Minerva | Olympiad | Avg. | ARC-c | GPQA* | MMLU-Pro | Avg. |
| Qwen-Base | 11.5 | 4.9 | 31.3 | 43.6 | 7.4 | 15.6 | 19.0 | 18.2 | 11.1 | 16.9 | 15.4 |
| Qwen-Instruct | 12.5 | 10.2 | 48.5 | 80.4 | 32.7 | 41.0 | 37.6 | 70.3 | 24.7 | 34.1 | 43.0 |
| *Previous Zero RLVR Methods* | | | | | | | | | | | |
| PRIME-Zero | 17.0 | 12.8 | 54.0 | 81.4 | 39.0 | 40.3 | 40.7 | 73.3 | 18.2 | 32.7 | 41.4 |
| Oat-Zero | 33.4 | 11.9 | 61.2 | 78.0 | 34.6 | 43.4 | 43.7 | 70.1 | 23.7 | 41.7 | 45.2 |
| GPG-Zero | 29.8 | 12.1 | 67.8 | 80.8 | 30.9 | 44.7 | 44.4 | 70.3 | 40.4 | 50.5 | 41.6 |
| RePO-Zero | 19.8 | 10.2 | 54.0 | 76.8 | 34.2 | 40.1 | 39.2 | 73.8 | 24.2 | 42.5 | 46.8 |
| *Zero RLVR with ExGRPO* | | | | | | | | | | | |
| On-Policy | 24.9 | 15.5 | 59.2 | 84.8 | 38.2 | 49.3 | 45.3 | 82.6 | 37.4 | 49.2 | 56.4 |
| **ExGRPO** | 31.6 | 18.7 | 66.3 | 87.4 | 36.0 | 50.1 | **48.3** | 84.7 | 37.4 | 52.9 | **58.3** |
| *Off-policy Learning Methods* | | | | | | | | | | | |
| SFT | 22.2 | 22.3 | 52.8 | 82.6 | 40.8 | 43.7 | 44.1 | 75.2 | 24.7 | 42.7 | 47.5 |
| SFT+RL | 25.8 | 23.1 | 62.7 | 87.2 | 39.7 | 50.4 | 48.2 | 72.4 | 24.2 | 37.7 | 44.8 |
| *Continual RLVR with ExGRPO* | | | | | | | | | | | |
| LUFFY | 29.4 | 23.1 | 65.6 | 87.6 | 37.5 | 57.2 | 50.1 | 80.5 | 39.9 | 53.0 | 57.8 |
| ↪ Continual LUFFY | 30.7 | 22.5 | 66.2 | 86.8 | 41.2 | 55.3 | 50.4 | 81.8 | 49.0 | 54.7 | **61.8** |
| ↪ On-Policy | 24.8 | 17.8 | 67.5 | 88.4 | 38.6 | 55.3 | 48.7 | 81.9 | 47.0 | 53.3 | 60.7 |
| ↪ **ExGRPO** | 32.3 | 25.7 | 65.6 | 87.6 | 40.1 | 57.0 | **51.4** | 83.6 | 42.4 | 54.5 | 60.2 |

AIME 2024/2025, AMC (Li et al., 2024), MATH-500 (Hendrycks et al., 2021), Minerva (Lewkowycz et al., 2022), and OlympiadBench (He et al., 2024). To assess generalization, we include three out-of-distribution tasks: ARC-c (Clark et al., 2018), GPQA-Diamond ("GPQA*"; Rein et al. 2024), and MMLU-Pro (Wang et al., 2024). For AIME, AMC benchmarks with a limited number of samples, we report *Avg@32* over 32 independent runs; for others, we use *Pass@1* metric, representing the proportion of solved problems on a single attempt. All evaluations use a sampling `temperature` of 0.6 and a `top-p` of 1.0, with answers extracted via the Oat-evaluator toolkit (Liu et al., 2025).

**RLVR Setups.** We use a rollout batch size of 128 and an update batch size of 64. During the rollout stage, we collect 8 sampled responses for on-policy questions and 7 for questions drawn from the replay buffer. The ratio $\rho$ of experience data in each mini-batch is set to 50%. For most models, the ExGRPO algorithm is activated only after the batch Pass@1 exceeds 35%. An exception is the Llama model, where this criterion is not applied due to the collapse of on-policy RLVR. Our experiments were conducted using the `verl` framework[3] on either 8×H100, with Math-Verify[4] as the reward model. More detailed training and validation settings are provided in Section E.1.

**Models and Baselines.** Our primary testbed for zero RLVR is the Qwen2.5-Math 7B model, and we also extend our evaluation to its 1.5B variant and other models, including Llama-3.1 8B (Base and Instruct) and Qwen2.5 7B Instruct models. Details of all baselines can be found in Section E.2.

## 5.2 MAIN RESULTS

**ExGRPO surpasses on-policy baselines on complex reasoning benchmarks.** Our main results in Table 1 show that ExGRPO consistently outperforms on-policy RLVR baselines across multiple benchmarks. When applied to the Qwen2.5-Math 7B model, ExGRPO delivers a +3.0 point gain over on-policy RLVR on math reasoning tasks. This advantage becomes more pronounced on challenging

---

[3]https://github.com/volcengine/verl/
[4]https://github.com/huggingface/Math-Verify

datasets such as AIME24/25, highlighting its effectiveness on difficult reasoning problems, and also holds on out-of-distribution benchmarks. A detailed comparison with the relevant replay-enhanced method in Section E.4 further shows that ExGRPO outperforms vanilla experiential RL baselines.

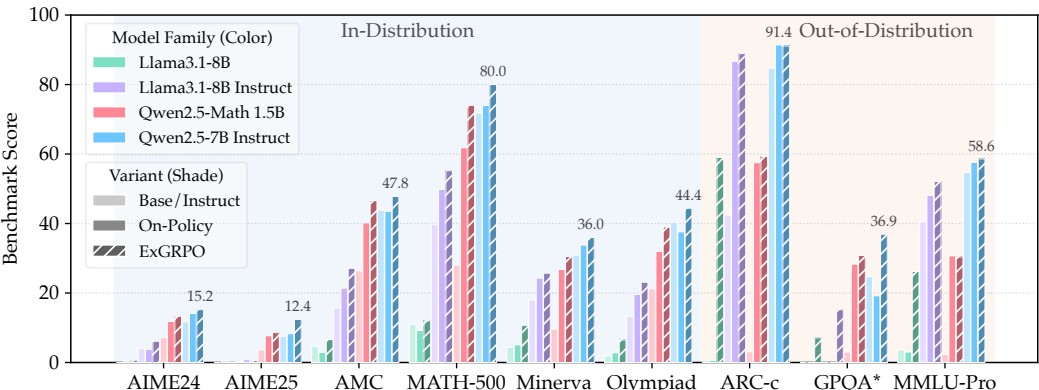

Figure 3: A comparison of benchmark performance for different backbone models and training variants, showing performance on both in-distribution and out-of-distribution tasks (cf. Section E.3).

**ExGRPO is robust across diverse model architectures and initializations.** We validate that the benefits of ExGRPO are not limited to a single model in Figure 3 and Appendix Table 5. ExGRPO has consistent improvements over baselines when applied to models of varying scales, including 1.5B and 8B parameter models, as well as models initialized from instruction-tuned checkpoints. For these models, ExGRPO achieves in-distribution improvements of up to +5.3 points on Qwen2.5-Math 1.5B Base and +4.1 points on Qwen2.5-7B Instruct. Notably, ExGRPO addresses a critical instability issue observed in on-policy RLVR, enabling successful training of Llama-3.1 base model on our data where the on-policy baseline method collapses. In a further experiment using the LUFFY model, which was trained with external off-policy data, we find that continual RLVR with model own experience yields greater performance gains than relying on external data. However, applying on-policy training to LUFFY still resulted in a degradation, highlighting the advantage of ExGRPO. We observe that ExGRPO underperforms on certain out-of-distribution tasks under *continual RLVR setting*, which we attribute to reduced on-policy exposure during joint mini-batch optimization with experience replay.

## 5.3 Extension to RL with Continuous Rewards

Beyond the binary-reward RLVR setup, extending ExGRPO to standard RLHF with continuous preference-model rewards is a vital next step. In this setting, we replace rollout success rates with the within-query reward variance computed over multiple rollouts. Reward variance serves as a natural signal of difficulty, and trajectory entropy remains applicable as a measure of reasoning validity.

For each query $q$ with $K$ rollouts, we compute the reward standard deviation $\sigma_q$. Since $\sigma_q$ has no intrinsic scale, we map it to a discrete difficulty bucket using relative, batch-wise normalization. Given all variances $\{\sigma_1, \ldots, \sigma_M\}$ within the current batch, we apply a linear min-max normalization $\tilde{\sigma}_q = \frac{\sigma_q - \sigma_{\min}}{\max(\sigma_{\max} - \sigma_{\min}, \varepsilon)}$, where $\varepsilon$ prevents degeneracy when the batch variance collapses. We then discretize $\tilde{\sigma}_q \in [0, 1]$ into $\{0, \ldots, K_{\mathrm{rollout}}\}$ via $\mathrm{round}(\tilde{\sigma}_q \cdot K_{\mathrm{rollout}})$. Each query is then assigned to its corresponding bucket and updated using the ExGRPO mechanism originally developed for the binary-correctness setting. Under this formulation, buckets still preserve the nature of discrete difficulty levels. As a result, ExGRPO's sampling mechanism, which was previously centered around medium success rates, now preferentially selects queries with moderate reward variance, i.e., those that are neither overly stable nor excessively noisy.

Table 2: Preliminary results extending ExGRPO to RL with continuous rewards.

| Method | ArenaHardV2 | IFEval | IFBench | Avg. |
|---|---|---|---|---|
| GRPO | 15.1 | 26.1 | 18.7 | 20.0 |
| ExGRPO | **16.4** | **27.9** | **20.7** | **21.7** |

These correspond to intermediate-difficulty queries where learning gains are often most substantial.

To validate feasibility, we conducted a study on 7k WildChat-IF samples (Zhao et al., 2025; Bhaskar et al., 2025) with on-policy RL training (2 epochs), using Qwen2.5-7B-Base as the base model and Skywork-Reward-Llama-3.1-8B-v0.2 (Liu et al., 2024) as the reward model. Results on chat and instruction-following benchmarks show a improvement of ExGRPO extension.

## 6 ANALYSIS AND DISCUSSION

To understand the effectiveness of ExGRPO, we investigate the following research questions: 1) How does experience replay influence training dynamics, especially for different models? 2) What is the impact of experience selection on overcoming performance bottlenecks? 3) How do the components of ExGRPO, e.g., experience management, contribute to overall performance?

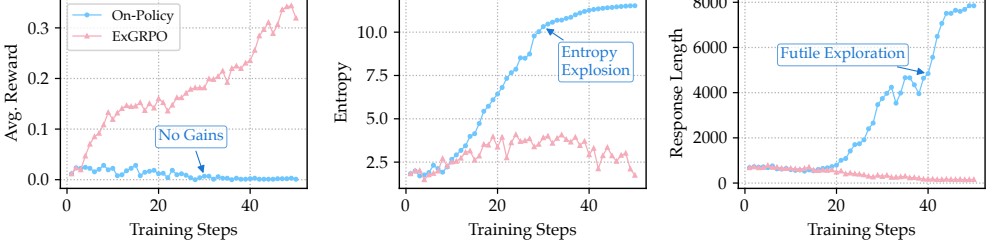

Figure 4: Learning dynamics of On-Policy vs. ExGRPO during training Llama-3.1 8B. ExGRPO stabilizes training and achieves higher rewards, while on-policy suffers from training collapse.

### 6.1 TRAINING DYNAMICS

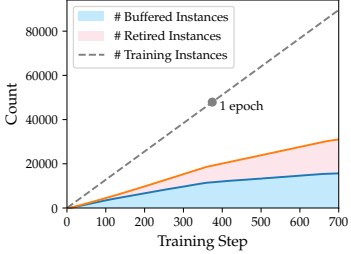

Figure 5: Dynamics of experience replay buffer and retried set.

**Experience replay helps stabilize training for weaker models.** Figure 4 depicts the training dynamics of the Llama3.1-8B base model. Due to its limited capacity, this model struggles to solve most problems, resulting in low rewards under on-policy training. Consequently, the exploration signal collapses, and the model finally encountered entropy explosion on challenging training data. In contrast, experience replay allows the model to exploit the "lucky hits" of correct solutions encountered early on. By replaying these successes, the model receives meaningful reward signals and is guided toward an exploration trajectory that matches its current capability, thereby stabilizing learning and preventing premature collapse.

**Experience replay improves data efficiency.** For stronger model, Figure 5 shows the evolution of the replay buffer and retired set. Their sum reflects the number of examples on which the model eventually succeeds. By the later stages, about half of the dataset achieves at least one successful trajectory (Pass@8), validating the promise of experiential learning: the model retains strong reasoning potential, and revisiting past successes enhances data utilization. The retired set further enhances efficiency: although initially small, it grows rapidly as capability improves and eventually matches the buffer in size. Retiring solved problems shifts training toward harder ones, reducing redundancy and explaining the slower buffer growth in later phases.

**The efficiency of experience utilization, rather than its sheer volume, is the primary driver of model performance.** Figure 6 illustrates the dynamics of the experience buffer and the retired set under different data conditions. When trained on the same amount of data, the absence of question selection mechanisms (e.g., *ExGRPO* vs. *w/o Ques. Sel.*) leads to growth of the retired set and a corresponding reduction in buffer size (primarily due to repeated exposure to easy questions). As a result, experience replay becomes inefficient, with high consumption of

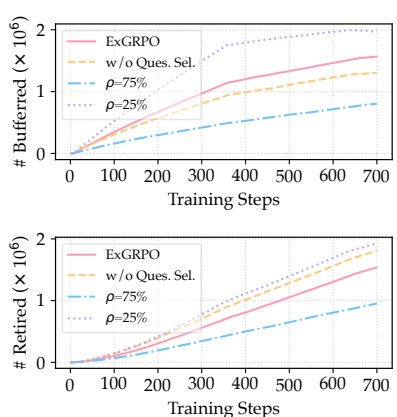

Figure 6: Dynamics of experience under different data conditions.

samples but low return in terms of performance gains. We further examine the scaling effects of the experience replay ratio, $\rho$. A high ratio ($\rho = 75\%$) leads to over-exploitation, stifling exploration and degrading performance below the on-policy baseline. Conversely, a low ratio ($\rho = 25\%$) prioritizes fresh on-policy learning; however, while it enlarges the experience buffer, the retired set grows only marginally. This indicates that accumulating more experiences does not guarantee improved capabilities. Ultimately, effective replay relies on balancing exploration with efficient sample reuse rather than simply maximizing buffer size, with optimal performance achieved at a balanced $\rho = 50\%$ (see results in Appendix Table 6).

## 6.2 ABLATION STUDIES

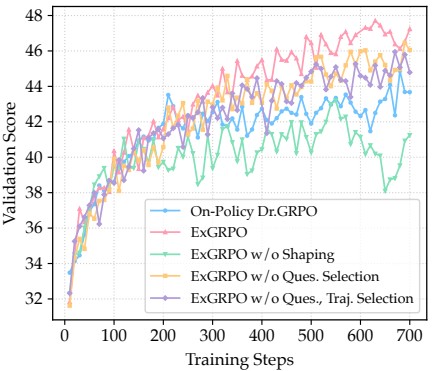

Figure 7: Comparison of validation performance of different ExGRPO variants.

**Empirical ablation supports ExGRPO's selection heuristics.** We conduct an ablation study to assess the contributions of **core** components of ExGRPO. Figure 7 shows performance dynamics of validation set (see Section E.1). For both question and trajectory selection, our analytically derived heuristics consistently outperform random selection, while random selection itself offers marginal gains over the on-policy baseline. Policy shaping mitigates this by ensuring exploitation does not hinder exploration. It is also noteworthy that applying shaping alone in on-policy RLVR yields no improvement (Yan et al., 2025), confirming that ExGRPO's gains cannot be attributed to naive shaping. The above results and additional ablations on test set (see Section E.4) provide empirical support for our design of ExGRPO.

**Sensitivity to difficulty sampling strategies.** Our method employs a probabilistic sampling scheme based on a Gaussian distribution $p \propto \mathcal{N}(\mathrm{Acc}(q); \mu = 0.5, \sigma)$. To assess the sensitivity of ExGRPO to the width of this distribution, we vary $\sigma \in \{0.5, 1.5\}$. A smaller $\sigma$ corresponds to a narrower medium-difficulty band, whereas a larger $\sigma$ approaches a near-uniform sampling distribution. We also evaluate a Hard-Biased variant by shifting the mean to $\mu = 0.25$, which emphasizes harder samples (25% accuracy) while reducing the chance of revisiting easier ones (75% accuracy). As shown in Table 3, the asymmetric sampling strategies yield inferior performance compared with the symmetric medium-difficulty focus ($\mu = 0.5$), confirming that balanced uncertainty regions provide the most effective signal.

Table 3: Sensitivity analysis of the Gaussian sampler with varying sampling focus and bias. **Bold** indicates the best results across settings for each column.

| Sampling Strategy | In-Distribution Performance | | | | | | | Out-of-Distribution Performance | | | |
|---|---|---|---|---|---|---|---|---|---|---|---|
| | AIME24 | AIME25 | AMC | MATH | Minerva | Olympiad | Avg. | ARC-c | GPQA* | MMLU-P | Avg. |
| On-Policy | 24.9 | 15.5 | 59.2 | 84.8 | 38.2 | 49.3 | 45.3 | 82.6 | 37.4 | 49.2 | 56.4 |
| **ExGRPO, ($\mu$=0.5, $\sigma$=1)** | **31.6** | 18.7 | **66.3** | **87.4** | 36.0 | **50.1** | **48.4** | **84.7** | 37.4 | **52.9** | **58.3** |
| Wider Focus ($\sigma$=1.5) | 22.0 | **19.2** | 62.3 | 85.4 | **38.2** | 49.9 | 46.2 | 81.7 | **44.4** | 48.6 | 58.2 |
| Narrower Focus ($\sigma$=0.5) | 24.9 | 17.4 | 64.9 | 86.6 | 34.2 | 49.0 | 46.2 | 72.9 | 40.4 | 51.1 | 54.8 |
| Hard-Biased ($\mu$=0.25) | 22.4 | 19.6 | 61.9 | 86.0 | 38.2 | 49.8 | 46.3 | 83.6 | 41.4 | 49.4 | 58.2 |

## 7 CONCLUSION

While experiential RL offers a promising approach to mitigate sample inefficiency in RLVR for large reasoning models, this area still lacks systematic exploration of experience value and management. In this work, we address this gap by first examining what constitutes a valuable reasoning experience, then introducing ExGRPO, a framework that strategically manages, selects and replays high-quality experiences. Experiments across multiple backbones and benchmarks show that ExGRPO consistently improves performance and stabilizes training where on-policy RLVR fails.

## ETHICS STATEMENT

This research follows the ICLR Code of Ethics. We have taken care to ensure that the datasets, methodologies, and model used in our experiments are ethically sound. Our work is primarily theoretical and empirical, focusing on optimizing reasoning models. As such, it does not involve sensitive personal data, or content likely to introduce societal bias, harm, or discrimination.

## REPRODUCIBILITY STATEMENT

We provide the experimental setups and hyperparameters in Section 5.1, with further details in Section E. The code and model weights are open-sourced.

## ACKNOWLEDGMENTS

We extend our gratitude to all the reviewers for their valuable feedback and suggestions. This work was supported by the Shanghai Artificial Intelligence Laboratory. This work was supported by the Science and Technology Development Fund of Macau SAR (Grant Nos. FDCT/0007/2024/AKP, EF2024-00185-FST), the UM and UMDF (Grant Nos. MYRG-GRG2024-00165-FST-UMDF, MYRG-GRG2025-00236-FST, SHMDF-AI/2026/001), and the National Natural Science Foundation of China (Grant No. 62266013).

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

# APPENDIX

APPENDIX CONTENTS

## A  LIMITATIONS

While our work demonstrates the significant benefits of principled experience management in RLVR, we acknowledge several limitations. First, since our research scope focused on RLVR, the evaluation tasks are verifiable problems, i.e., mathematical and general reasoning benchmarks where trajectories can be more easily verified as correct or incorrect. The applicability of our correctness-based bucketing strategy to more open-ended tasks (e.g., creative writing), where rewards are often subjective and dense, remains an open question. Second, our framework's definition of a valuable experience is based on heuristics that, while powerful, might be incomplete. ExGRPO might neglect the learning potential of "valuable failures" where some incorrect paths that contain useful training signals (Zhu et al., 2025). This focus on exploitation could risk premature convergence in some scenarios. Finally, while ExGRPO is built upon a relative policy optimization objective, its interaction with other families

of RL algorithms has not been explored. Future work includes extending our method to multi-modal reasoning and agentic reinforcement learning.

## B  THE USE OF LARGE LANGUAGE MODELS (LLMs)

We utilize large language models to assist with proofreading, and polishing the text of this paper. The generated responses are treated as decision references and are not adopted wholesale. Importantly, all scientific contributions, conceptual developments, experimental designs, and final decisions were made by human researchers. The outputs from LLMs were treated as references or drafts, subject to human review, modification, and validation.

## C  ExGRPO ALGORITHM

---

**Algorithm 1** ExGRPO: Experiential Group Relative Policy Optimization

---

**Require:** Dataset $\mathcal{D}$, batch size $B$, experience ratio $\rho \in [0, 1]$, rollout trails $K$, reward model $R(\cdot)$.
  1: **Initialize:** Policy model $\pi_\theta$, experience replay buffer $\mathcal{E} \leftarrow \emptyset$, retired set $\mathcal{S} \leftarrow \emptyset$.
  2: **for** each training step **do**
  3:     ▷ *Phase 1: Experience Management.*
  4:     **if** $|\mathcal{E}| > 0$ **then**
  5:         Partition $\mathcal{E}$ to $k$ buckets $\mathcal{U}_k$ by *last* rollout correctness $\mathrm{Acc} = \frac{1}{K} \sum_{k=1}^{K} \mathbf{1}[r_k = 1]$.
  6:         Obtain sampling probability: $p_k = \mathcal{N}(k/K; 0.5, \sigma = 1)$ for nonempty $\mathcal{U}_k$
  7:         Calculate experience sampling amount: $n \leftarrow \min(\lfloor \rho B \rfloor, |\mathcal{E}|)$
  8:         Sample $n$ experience: $\{e_1, \ldots, e_n\} \sim \mathsf{BukcetSample}(\mathcal{U}, \{p_k\})$          ▷ cf. Algorithm 2.
  9:         **for** each experience $e \in \{e_1, \ldots, e_n\}$ **do**          ▷ cf. $\mathcal{E}: q^* \mapsto \{o^*\}$
 10:             **for** each experience trajectory $o_i \in e$ **do**
 11:                 Compute trajectory entropy under $\pi_\theta$: $H(o_i; \pi_\theta) = -\frac{1}{|o_i|} \sum_t \log \pi_\theta(o_i^t | q^*, o_i^{<t})$
 12:             **end for**
 13:             Select the trajectory with the lowest entropy: $o^* \leftarrow \arg\min_{e \in \mathcal{E}} H(o_i; \pi_\theta)$
 14:         **end for**
 15:         Experiential policy optimization data $\mathcal{B}_{\mathrm{exp}} \leftarrow \{(q^*, o^*)\}^n$
 16:         On-policy optimization data $\mathcal{B}_{\mathrm{on}} \leftarrow \{q\}^{B-n} \sim \mathcal{D}$
 17:     **else**
 18:         $\mathcal{B}_{\mathrm{exp}} \leftarrow \emptyset$, $\mathcal{B}_{\mathrm{on}} \leftarrow \{q\}^B \sim \mathcal{D}$
 19:     **end if**
 20:     ▷ *Phase 2: Experiential Policy Optimization.*
 21:     Construct batch data: $\mathcal{B} \leftarrow \mathcal{B}_{\mathrm{exp}} \cup \mathcal{B}_{\mathrm{on}}$
 22:     **for** each sample $q \in \mathcal{B}$ **do**
 23:         Generate rollouts: $\{o_1, \ldots, o_K\} \sim \pi_\theta(\cdot | q), \{o_1, \ldots, o_{K-1}\} \sim \pi_\theta(\cdot | q^*)$
 24:         Compute rewards and success: $\{r_j\}_{j=1}^{K} = \{R(q, o_j)\}_{j=1}^{K}; s = \sum_{j=1}^{K} \mathbb{I}[r_j = 1]$
 25:         **if** $s = K$ **then**          ▷ All successful
 26:             $\mathcal{S} \leftarrow \mathcal{S} \cup \{q\}$          ▷ Add to retired set
 27:         **else if** $s < K$ **then**          ▷ Partial success
 28:             Record experience: $\mathcal{E}[q] \cup \{o_j \mid r(o_j) = 1\}$
 29:         **end if**
 30:     **end for**
 31:     Compute advantage and update policy model with Eq. 4.
 32:     Remove well-learned samples: $\mathcal{E} \leftarrow \mathcal{E} \setminus \{q : q \in \mathcal{S}\}$
 33: **end for**
 34: **return** Optimized policy model $\hat{\pi}_\theta$

---

**Overview of ExGRPO (Algorithm 1).**  ExGRPO organizes training into two phases: *experience management* and *experiential policy optimization*. In first phase, the replay buffer $\mathcal{E}$ is partitioned into buckets based on the most recent rollout correctness $\mathrm{Acc}(q)$, and a Gaussian weighting centered at 0.5 biases sampling toward medium-difficulty problems. From these buckets, $n = \min(\lfloor \rho B \rfloor, |\mathcal{E}|)$

experiences are drawn, and for each, the trajectory with the lowest entropy under the current policy is selected as $o^*$. These anchors form the experiential sub-batch $\mathcal{B}_{\text{exp}}$, while the remaining slots are filled with on-policy samples $\mathcal{B}_{\text{on}}$. In the second phase, each batch item generates $K$ rollouts (or $K-1$ plus the past trajectory), rewards are computed, and prompts achieving full success are retired, while partially successful ones refresh $\mathcal{E}$. The policy is updated using objective in Eq. 4.

**Details of Bucketed Sampling (Algorithm 2).** This sampler first normalizes probabilities $\mathbf{p}$ over non-empty buckets, then draws bucket counts $\mathbf{c}$ using a sequential-binomial implementation of the multinomial distribution, ensuring $\sum_k c_k = n$ where $n$ is the number of samples to be drawn. Within each bucket, $c_k$ items are sampled uniformly without replacement. This preserves the intended inter-bucket bias (e.g., Gaussian weighting by accuracy) while remaining unbiased within buckets. The method assumes $c_k \leq |\mathcal{U}_k|$ for feasibility. In practice, to handle rare edge cases where a bucket becomes empty, clipping and redistributing the sampling counts will be applied in such scenarios.

---

**Algorithm 2** BucketSample(). Bucketed Multinomial & Within-Bucket Uniform Sampling

---

**Require:** Valid buckets $\mathcal{U} = \{(\text{Acc}, \{e\})\}_{k=1}^{K}$; probabilities $\mathbf{p} \in [0,1]^K$; sample size $n \in \mathbb{N}$;
**Ensure:** probabilities $\mathbf{p} \in [0,1]^K$ with $\sum_{k=1}^{K} p_k = 1$
 1: **function** MULTINOMIAL($n, \mathbf{p}$)
 2:     *# Note: sample $X \in \mathbb{N}^d$ such that $\sum_{i=1}^{d} X_i = n$*
 3:     Initialize $X \leftarrow (0, \ldots, 0) \in \mathbb{N}^d$, $m \leftarrow n$
 4:     **for** $i = 1$ to $d - 1$ **do**
 5:         Draw $X_i \sim \text{Binomial}\left(m, \frac{p_i}{1-\sum_{j<i} p_j}\right)$
 6:         Update $m \leftarrow m - X_i$
 7:     **end for**
 8:     **return** collection of $X$
 9: **end function**

10: *# Note: sampling within a bucket is uniform* without *replacement; requires $c_k \leq |\mathcal{U}_{\text{Acc}}|$ for all $k$.*
11: $\mathbf{c} \sim \text{Multinomial}(n, \mathbf{p})$             ▷ Draw all bucket counts in one shot
12: sampled_items $\leftarrow \{\}$
13: **for** $k \leftarrow 1$ **to** $K$ **do**
14:     $S \leftarrow \text{UniformSample}(\mathcal{U}_k, c_k)$            ▷ Without replacement.
15:     sampled_items$[\mathcal{U}_k] \leftarrow S$
16: **end for**
17: **return** (sampled_items, [])

---

**Complexity.** Experience management in ExGRPO is efficient. Per step, ExGRPO performs $O(BK)$ reward evaluations and $O(BK)$ likelihood computations. Bucket partitioning and sampling are $O(|\mathcal{E}|)$ for bucketization (amortized) and $O(K + n)$ for drawing counts and items.

**Correctness and Constraints.** The procedure assumes $c_k \leq |\mathcal{U}_k|$ for all nonempty buckets. In practice, this is satisfied with high probability when (i) the support of $\mathbf{p}$ excludes tiny buckets or (ii) we clip $c_k$ to $|\mathcal{U}_k|$ and redistribute any deficit to remaining buckets. We adopt the simple feasibility requirement as stated in Algorithm 2 for clarity.

**Practical Notes.** We renormalize $\mathbf{p}$ over *nonempty* buckets before sampling; this avoids assigning mass to empty buckets. The scheme is streaming-friendly and yields $O(K)$ arithmetic plus the cost of within-bucket draws. When used in ExGRPO, setting $\mathbf{p} \propto \mathcal{N}(k/K; \mu, \sigma)$ provides a tunable curriculum over the rollout correctness spectrum. In this paper, we consistently set $\mu = 0.5, \sigma = 1$ to align with our analytical findings in Section 3.

# D  THEORETICAL ANALYSIS OF ExGRPO

## D.1  RECAP AND FORMULATION

Let $q \sim \mathcal{D}$ be a query. The rollout (reference) policy is $\pi_{\theta_{\text{old}}}$, the current policy is $\pi_\theta$, and the experiential policy (that produced a replayed trajectory) is $\pi_{\theta_{\text{past}}}$. For each $q$, form a group $\mathcal{G}_q = \{o_i\}_{i=1}^K$ with $o_i \sim \pi_{\theta_{\text{old}}}(\cdot \mid q)$ and use GRPO's within-group standardization

$$\widehat{A}(o_i, \mathcal{G}_q) = \frac{r(q, o_i) - \mu_{\mathcal{G}_q}}{\sigma_{\mathcal{G}_q}}, \quad \mu_{\mathcal{G}_q} = \tfrac{1}{K} \sum_{i=1}^K r(q, o_i), \ \ \sigma_{\mathcal{G}_q} = \text{Std}\big(\{r(q, o_i)\}_{i=1}^K\big), \tag{5}$$

Based on the practice we detailed in the main body of paper, we follow the simplifications advocated by Dr.GRPO (Liu et al., 2025), removing length and standard deviation normalization. For mixed-policy optimization, we also remove clipping and use policy shaping.

ExGRPO mixes (i) on-policy groups $\mathcal{G}_q$ and (ii) *mixed* groups $\mathcal{G}_{q^*} = \{o^*\} \cup \{o_i\}_{i=1}^{K-1}$ for $q^*$ sampled from an experience buffer, where $o^* \sim \pi_{\theta_{\text{past}}}(\cdot \mid q^*)$ and the $K-1$ fresh rollouts come from $\pi_{\theta_{\text{old}}}(\cdot \mid q^*)$. The ExGRPO mini-batch objective (Eq. 4) mixes on-policy and experiential (replayed) contributions with weight $\rho \in [0, 1)$.

Per-token importance ratios for a trajectory $o$ comparing policy $\pi_\theta$ to policy $\pi_{\theta'}$ (i.e., $\pi_{\theta_{\text{old}}}$ or $\pi_{\theta_{\text{past}}}$) are:

$$w_t(o_i; \theta, \theta') := \frac{\pi_\theta(o_{i,t} \mid q, o_{i,<t})}{\pi_{\theta'}(o_{i,t} \mid q, o_{i,<t})}, \qquad W(o_i; \theta, \theta') := \prod_{t=1}^{|o|} w_t(o_{i,t}; \theta, \theta'). \tag{6}$$

We will sometimes write $w(\cdot)$ or $W(\cdot)$ for brevity. For the replayed item $o^*$, ExGRPO uses a bounded *policy-shaping* transform $f(w) = \frac{w}{w+\beta}$ with $\beta > 0$ instead of hard clipping; this is analogous to truncated/saturated importance ratios in ACER and V-trace, known to control variance at small bias (Wang et al., 2016; Espeholt et al., 2018). We assume bounded rewards, absolute continuity of policies (mutual support overlap), and an auxiliary KL control as in Schulman et al. (2015; 2017).

**Assumptions.** For formal statements we will use the following standard assumptions where invoked:

*A1. Support assumption.* For any trajectory $o$ produced by a past policy used in replay, $\pi_\theta(o_t \mid \cdot) > 0$ whenever $\pi_{\theta_{\text{past}}}(o_t \mid \cdot) > 0$ (i.e., target policy has support containing past policy).

*A2. Bounded second moment.* For any random variable $X$ under consideration (rewards, advantages), second moments exist and are finite. Formally, for any policy model $\pi_\theta$ and query distribution $\mathcal{D}$,

$$\mathbb{E}_{q \sim \mathcal{D}, \, o \sim \pi_\theta(\cdot|q)}\big[X^2\big] < \infty \tag{7}$$

so finiteness of $\mathbb{E}[X^2]$ guarantees that $\text{Var}(X) = \mathbb{E}[X^2] - \big(\mathbb{E}[X]\big)^2$ is well-defined.

*A3. Bounded importance ratios.* Assume finite trajectory length, there exists $M \geq 1$ such that for all relevant trajectories and timesteps,

$$\sup_{o,t} \frac{\pi_\theta(o_t \mid \cdot)}{\pi_{\theta_{\text{past}}}(o_t \mid \cdot)} \leq m, \quad \text{and hence} \quad \sup_o W(o; \theta, \theta_{\text{past}}) \leq M := m^{|o|}. \tag{8}$$

## D.2  UNBIASEDNESS OF EXPERIENTIAL GRADIENT

We show that, when the replayed trajectory is corrected by the exact per-token importance weight $W(o^*; \theta, \theta_{\text{past}})$, the contribution of that trajectory yields an unbiased estimate of the corresponding on-policy term (no bias is introduced by using replay plus exact importance reweighting), even when the advantage $\widehat{A}$ depends on the full mixed group $\mathcal{G}$.

**Theorem 1** (Unbiasedness). *Under Assumption A1, let $\mathcal{G}_{q^*} = \{o^*\} \cup \{o_i\}_{i=1}^{K-1}$ be a mixed group where $o^*$ was sampled from $\pi_{\theta_{\text{past}}}$ and $\{o_i\}$ were sampled from $\pi_{\theta_{\text{old}}}$. For any measurable function*

*$g$ of a trajectory and its group (e.g. $g(o, \mathcal{G}) = \sum_t \nabla_\theta \log \pi_\theta(o_t | q, o_{<t}) \cdot \widehat{A}(o, \mathcal{G})$), the importance-weighted expectation equals the on-policy expectation:*

$$\mathbb{E}_{o^* \sim \pi_{\theta_{past}}} \left[ W(o^*; \theta, \theta_{past}) \, g(o^*, \mathcal{G}_{q^*}) \mid \{o_i\}_{i=1}^{K-1} \right] = \mathbb{E}_{\tilde{o} \sim \pi_\theta} \left[ g(\tilde{o}, \tilde{\mathcal{G}}) \mid \{o_i\}_{i=1}^{K-1} \right],$$

*where the right hand side is the expectation with $o^*$ replaced by sampling $\tilde{o} \sim \pi_\theta$ and forming group $\tilde{\mathcal{G}}$ accordingly.*

*Proof Sketch.* Condition on the other group elements $\{o_i\}_{i=1}^{K-1}$, $\{o_i\}_{i=1}^{K-1}$ is implicit as the sampling of $o^*$ is independent of $\{o_i\}_{i=1}^{K-1}$, thus we have

$$\mathbb{E}_{o^* \sim \pi_{\theta_{\text{past}}}} \left[ W(o^*; \theta, \theta_{\text{past}}) \, g(o^*, \mathcal{G}_{q^*}) \mid \{o_i\}_{i=1}^{K-1} \right] = \mathbb{E}_{o^* \sim \pi_{\theta_{\text{past}}}} \left[ W(o^*; \theta, \theta_{\text{past}}) \, g(o^*, \mathcal{G}_{q^*}) \right] \quad (9)$$

Further, consider the expectation over $o^*$ drawn from $\pi_{\theta_{\text{past}}}$. By the definition of expectation, we sum over all possible trajectories (which we denote by $o^*$) weighted by their probability under $\pi_{\theta_{\text{past}}}$:

$$\begin{aligned}
\mathbb{E}_{o^* \sim \pi_{\theta_{\text{past}}}} \left[ W(o^*; \theta, \theta_{\text{past}}) \, g(o^*, \mathcal{G}_{q^*}) \right] &= \sum_{o^*} \pi_{\theta_{\text{past}}}(o^*) \left[ W(o^*; \theta, \theta_{\text{past}}) \, g(o^*, \mathcal{G}_{q^*}) \right] \\
&= \sum_{o^*} \pi_{\theta_{\text{past}}}(o^*) \frac{\pi_\theta(o^*)}{\pi_{\theta_{\text{past}}}(o^*)} g(o^*, \mathcal{G}_{q^*}) \\
&= \sum_{o^*} \pi_\theta(o^*) g(o^*, \mathcal{G}_{q^*}) \\
&= \mathbb{E}_{\tilde{o} \sim \pi_\theta} \left[ g(\tilde{o}, \mathcal{G}) \right].
\end{aligned} \quad (10)$$

The key is that $g(\cdot, \mathcal{G})$ is an arbitrary measurable function of a trajectory and the group, and the algebra above does not require $g$ to be independent of the group statistics (the sum is over the trajectory variable and the group members are treated as constants in the inner sum). This yields the stated identity. Thus, we have shown that the importance-weighted experiential term is an unbiased estimator of the on-policy term, conditioned on the rest of the group members. $\square$

**Remark.** The theorem shows that, with exact importance weights, replay introduces no bias even when the advantage is computed using group-dependent normalization ($\mu_{\mathcal{G}}$, $\sigma_{\mathcal{G}}$). In practice exact per-token ratios are used (product over timesteps), and the identity holds under A1.

### D.3    VARIANCE DECOMPOSITION AND BOUNDS

Let $G_{\text{exp}}$ denotes the random contribution from an experience-mixed group that includes one replayed trajectory corrected by importance sampling as:

$$G_{\text{exp}} = \frac{1}{K} \left( W(o^*; \theta, \theta_{\text{past}}) \, U(o^*, \mathcal{G}) + \sum_{i=1}^{K-1} U(o_i, \mathcal{G}) \right), \quad (11)$$

where $U(o, \mathcal{G})$ denotes the per-trajectory *unweighted* gradient contribution (e.g. $\sum_t \nabla_\theta \log \pi_\theta(\cdot) \, \widehat{A}$).

Importantly, in our implementation of GRPO the group standard-deviation term is omitted (we only mean-center advantages), so the $U(\cdot, \mathcal{G})$ terms still depend on the group mean $\mu_{\mathcal{G}}$ but not on $\sigma_{\mathcal{G}}$. This reduces but does not eliminate the statistical coupling between group members.

Below we give a general variance bound that does not assume independence among group members, and then a tighter bound under a weak-independence assumption that is often approximately valid in practice (e.g., large $K$).

**Proposition 2** (Variance upper bound for experiential term). *Under Assumption A2 (finite second moments) and A3 (bounded trajectory importance ratios), the experiential variance satisfies the following bounds.*

1. *(General, no independence) Without assuming independence between $U(o^*, \mathcal{G})$ and the other group contributions, we have the conservative inequality*

$$\text{Var}\left(G_{\text{exp}}\right) \leq \frac{2}{K^2}\Big(\mathbb{E}\big[W^2 U^2\big] + (K-1)^2 \mathbb{E}\big[U^2\big]\Big). \tag{A}$$

*If in addition $W$ is uniformly bounded by $M$ (Assumption A3), then*

$$\text{Var}\left(G_{\text{exp}}\right) \leq \frac{2}{K^2}\Big(M^2 \mathbb{E}[U^2] + (K-1)^2 \mathbb{E}[U^2]\Big) = \frac{2\big(M^2 + (K-1)^2\big)}{K^2} \mathbb{E}[U^2]. \tag{A'}$$

2. *(Tighter bound under weak/approximate independence) If the within-group unweighted contributions $\{U(o_i, \mathcal{G})\}_{i=1}^{K-1}$ are pairwise uncorrelated and uncorrelated with $W(o^*)U(o^*, \mathcal{G})$ (e.g., approximate when $K$ is large), then*

$$\text{Var}\left(G_{\text{exp}}\right) \leq \frac{2}{K^2}\Big(\mathbb{E}\big[W^2 U^2\big] + (K-1)\mathbb{E}[U^2]\Big). \tag{B}$$

*With the bound $W \leq M$ this yields*

$$\text{Var}\left(G_{\text{exp}}\right) \leq \frac{2\big(M^2 + (K-1)\big)}{K^2} \mathbb{E}[U^2]. \tag{B'}$$

*Proof Sketch.* Write $S = \sum_{i=1}^{K-1} U(o_i, \mathcal{G})$. Then

$$\text{Var}\left(G_{\text{exp}}\right) = \frac{1}{K^2}\text{Var}\left(WU + S\right) \leq \frac{1}{K^2}\mathbb{E}\big[(WU + S)^2\big] \tag{12}$$

using $\text{Var}(X) \leq \mathbb{E}[X^2]$. Expanding and applying *Cauchy–Schwarz* to bound cross terms yields

$$\text{Var}\left(G_{\text{exp}}\right) \leq \frac{1}{K^2}\Big(\mathbb{E}[W^2 U^2] + \mathbb{E}[S^2] + 2\sqrt{\mathbb{E}[W^2 U^2]\,\mathbb{E}[S^2]}\Big). \tag{13}$$

Using $(a + b + 2\sqrt{ab}) \leq 2(a + b)$ and $\mathbb{E}[S^2] \leq (K-1)^2\mathbb{E}[U^2]$ (which follows without independence by bounding pairwise covariances by second moments) gives the general bound (A). Plugging $W^2 \leq M^2$ yields (A').

If one further assumes pairwise uncorrelatedness of the $U$-terms and zero covariance with $WU$, then $\mathbb{E}[S^2] = (K-1)\mathbb{E}[U^2]$ and the cross terms drop, recovering the tighter bound (B) and its specialization (B'). $\qquad\square$

**Practical Remark.** Removing the within-group standard-deviation from GRPO reduces the coupling introduced by a stochastic denominator (the $\sigma_\mathcal{G}$-term). This typically lowers $\mathbb{E}[U^2]$ compared to the case with standard deviation normalization (because the denominator's variability can strongly amplify per-trajectory contributions), which in turn reduces all bounds above.

Based on the upper bound analysis of variance, an important insight is that controlling the source of experience policy is crucial for reducing the value of $M$, and consequently tightening the variance bound. In practice, techniques such as low-entropy selection and importance sampling correction contribute to this goal. Specifically, we use the average conditional entropy of the trajectory as pickup metric, and minimizing this metric encourages trajectories from the typical set, which facilitates controlling the importance sampling term and reduces the discrepancy between $\pi_\theta$ and $\pi_{\theta_{\text{old}}}$.

### D.4 FINAL REMARKS

**Policy shaping via $f(w) = \frac{w}{w+\beta}$ and removing clip.** As mentioned in Section 4.2, we remove the CLIP term and replace it with a policy-shaping function to enable better off-policy experience exploitation. Rather than relying on empirical validation from prior work (use expert trajectory as off-policy), we also provide a discussion of this design choice used in experiential optimization here. The policy shaping transform replaces $w$ by $f(w)$ for the replayed trajectory. Its key properties are:

- $f$ is monotone increasing in $w$.

- $0 \le f(w) < 1$ for all $w \ge 0$, and $f(w) \to 1$ as $w \to \infty$.

- For small $w$, $f(w) \approx w/\beta$ (which can amplify very small ratios when $\beta < 1$); for large $w$, $f(w)$ is bounded by 1 and thus damps extremely large ratios.

Thus $f$ reduces variance contribution of large importance weights while introducing bias (since $\mathbb{E}[f(W)] \neq 1$ generally). Combined with group normalization, medium-difficulty question sampling and low-entropy trajectory, policy shaping often suffices to control variance so that GRPO clipping can be relaxed or removed in practice.

**Summary and Takeaways.** We have shown that exact trajectory-level importance weighting guarantees unbiasedness of experiential contributions, even when advantages are computed from mixed groups (Theorem 1). The variance of experiential gradients is governed by the second moment $\mathbb{E}[W^2 U^2]$, so bounding or controlling importance sampling term is critical (Proposition 2).

Finally, the smooth policy-shaping transform $f(w) = \frac{w}{w+\beta}$ offers a principled bias–variance tradeoff: it suppresses extreme importance weights while preserving informative contributions that would be truncated by hard clipping. In combination with within-group normalization, this explains why ExGRPO can remain stable without clipping while still benefiting from replayed trajectories.

# E    SUPPLEMENTARY MATERIALS OF EXPERIMENTS

## E.1    DETAILED TRAINING SETTINGS

During RLVR training, the rollout generation uses a `temperature` of $1.0$. To reduce training time and monitor training dynamics, we used a validation set by sampling 2.2k instances from the test set. Model performance was evaluated every 10 steps to better track training progress, without applying early stopping. Qwen2.5-Math 7B backbone was trained for 700 steps, while the models in our extension experiments were trained for 500 steps. Following the methodology of LUFFY (Yan et al., 2025), we employed a consistent prompt template across all models during training, as detailed in Section G.1. An exception was made for the Llama-3.1 8B base model, for which a simplified prompt was used to accommodate its limited capabilities. Additional training hyper-parameters are provided in Table 4.

## E.2    BASELINES

For the zero RLVR using the Qwen2.5-Math 7B model (Yang et al., 2024), we compare our approach against not only the on-policy Dr.GRPO baseline but also other RLVR methods:

- **PRIME-Zero** (Cui et al., 2025a): PRIME-Zero is trained with PRIME, an online RL framework that leverages implicit process rewards (Yuan et al., 2024) to enhance reasoning beyond imitation or distillation. In PRIME, both the policy and implicit PRM are initialized from the SFT model; the policy generates rollouts, which are scored by the implicit PRM and an outcome verifier. The policy is then updated using a combination of outcome and process rewards.

- **Oat-Zero** (Liu et al., 2025): The original Dr.GRPO implementations discard the standard deviation in advantage computation and omit token-level normalization in the policy loss.

- **GPG-Zero** (Chu et al., 2025): GPG directly integrates group-based decision dynamics into standard policy gradients, simplifying training and greatly reducing computation without loss of reasoning quality.

- **RePO-Zero** (Li et al., 2025): Replay-enhanced Policy Optimization (RePO) employs an online experience replay mechanism by collecting early on-policy rollouts. However, it revisits them asynchronously and does not incorporate experience management. We further compare ExGRPO with RePO under the same data and training settings used by RePO, as detailed in Section E.4.

Table 4: RLVR training hyperparameters used in our experiments. All sequence lengths are measured using each model's own tokenizer.

| Module | Parameter | Value | Description |
|---|---|---|---|
| Data | data.train_batch_size | 128 | Global training batch size per optimization step. |
| | data.val_batch_size | 512 | Batch size used for validation. |
| | data.max_prompt_length | 1024 | Maximum input length. |
| | data.max_response_length | 8192 | Maximum response length. |
| Actor | actor_rollout_ref.actor.optim.lr | $1 \times 10^{-6}$ | Learning rate for the actor optimizer. |
| | actor_rollout_ref.actor.ppo_mini_batch_size | 64 | - |
| | actor_rollout_ref.actor.ppo_micro_batch_size | 64 | - |
| | actor_rollout_ref.actor.entropy_coeff | 0.001 | - |
| | actor_rollout_ref.actor.loss_remove_token_mean | True | Remove token-wise mean term from the loss. |
| | actor_rollout_ref.actor.loss_remove_clip | True | Disable the clipping term in the loss. |
| Rollout | actor_rollout_ref.rollout.engine | vllm | Inference/rollout engine. |
| | actor_rollout_ref.rollout.temperature | 1.0 | Sampling temperature during training rollouts. |
| | actor_rollout_ref.rollout.val_temperature | 0.6 | Sampling temperature during validation. |
| | actor_rollout_ref.rollout.gpu_memory_utilization | 0.80 | Fraction of GPU memory to utilize for the rollout engine. |
| | actor_rollout_ref.rollout.n | 8 | Number of rollouts per prompt during RLVR. |
| Trainer | trainer.critic_warmup | 0 | Number of warmup steps (0 disables warmup). |
| | trainer.training_steps | 700/500 | Number of training steps. 700 steps for Qwen2.5-Math-7B, 500 steps for other models. |

For the continual RLVR setting, we apply ExGRPO on a strong backbone model, LUFFY (Yan et al., 2025), which integrates Deepseek-R1 (Guo et al., 2025) trajectories in RL and substantially boosts reasoning performance. We further compare ExGRPO against other reproduced off-policy learning strategies, including:

- **SFT**: Supervised fine-tuning directly on the R1 trajectories from OpenR1 Face (2025).

- **SFT+RL**: A two-stage approach involving SFT on R1 trajectories followed by on-policy RL on the same problem set.

- **Continual LUFFY**: A direct continual off-policy optimization on the R1 trajectories using the LUFFY algorithm.

Table 5: Overall in-distribution and out-of-distribution performance based on different backbone models. **Bold** indicates the best results within a comparable group.

| Model | In-Distribution Performance | | | | | | | Out-of-Distribution Performance | | | |
|---|---|---|---|---|---|---|---|---|---|---|---|
| | AIME24 | AIME25 | AMC | MATH-500 | Minerva | Olympiad | Avg. | ARC-c | GPQA* | MMLU-Pro | Avg. |
| *Qwen2.5-Math 1.5B* | | | | | | | | | | | |
| Base | 7.2 | 3.6 | 26.4 | 28.0 | 9.6 | 21.2 | 16.0 | 3.1 | 3.0 | 2.3 | 2.8 |
| On-Policy | 11.8 | 7.7 | 40.2 | 61.8 | 26.8 | 32.0 | 30.0 | 57.5 | 28.3 | 30.7 | 38.9 |
| **ExGRPO** | 13.3 | 8.6 | 46.6 | 74.0 | 30.5 | 39.0 | **35.3** | 59.3 | 30.8 | 30.4 | **40.2** |
| *Qwen2.5-7B Instruct* | | | | | | | | | | | |
| Instruct | 11.7 | 7.5 | 43.8 | 71.8 | 30.9 | 40.4 | 34.4 | 84.7 | 24.7 | 54.7 | 54.7 |
| On-Policy | 14.1 | 8.3 | 43.5 | 74.0 | 33.8 | 37.6 | 35.2 | 91.4 | 19.2 | 57.6 | 56.0 |
| **ExGRPO** | 15.2 | 12.4 | 47.8 | 80.0 | 36.0 | 44.4 | **39.3** | 91.1 | 36.9 | 58.6 | **62.2** |
| *Llama3.1-8B* | | | | | | | | | | | |
| Base | 0.0 | 0.0 | 4.7 | 11.0 | 4.4 | 1.9 | 3.7 | 0.2 | 0.0 | 3.6 | 1.3 |
| On-Policy | 0.4 | 0.4 | 2.9 | 9.2 | 5.1 | 2.8 | 3.4 | 0.6 | 0.0 | 3.0 | 1.2 |
| **ExGRPO** | 0.7 | 0.1 | 6.6 | 12.0 | 10.7 | 6.5 | **6.1** | 59.0 | 7.1 | 26.2 | **30.8** |
| *Llama3.1-8B Instruct* | | | | | | | | | | | |
| Instruct | 4.0 | 0.3 | 15.7 | 39.8 | 18.0 | 13.3 | 15.2 | 42.4 | 0.0 | 40.5 | 27.6 |
| On-Policy | 3.8 | 0.8 | 21.4 | 49.8 | 24.3 | 19.6 | 19.9 | 86.7 | 0.0 | 48.1 | 44.9 |
| **ExGRPO** | 6.1 | 0.6 | 27.1 | 55.2 | 25.7 | 23.1 | **23.0** | 88.9 | 15.2 | 52.1 | **52.0** |

### E.3 RESULTS OF MODEL EXTENSION

The empirical results in Table 5 provide strong evidence that ExGRPO offers significant and consistent performance gains over both the original models and a strong On-Policy baseline, demonstrating its efficacy and robustness across diverse model architectures and initial tuning states. The evaluation spans multiple model families and sizes, including both base (Qwen2.5-Math 1.5B, Llama3.1-8B) and instruction-tuned variants (Qwen2.5-7B Instruct, Llama3.1-8B Instruct), to demonstrate the broad applicability of our method. The results consistently demonstrate the superiority of ExGRPO over the On-Policy method across all tested configurations. The advantage of ExGRPO is particularly pronounced when fine-tuning from base models. On Llama3.1-8B, the On-Policy method shows negligible gains over the base model, whereas ExGRPO provides a substantial boost, improving the average ID performance from 3.7 to 6.1 and, most notably, the average OOD performance from a mere 1.3 to 30.8. This highlights ExGRPO's effectiveness in robustly enhancing model capabilities, especially in scenarios where standard on-policy fine-tuning struggles to yield improvements.

### E.4 ABLATION AND ADDITIONAL RESULTS

To dissect the contribution of each key component within our proposed ExGRPO framework, we conduct a series of ablation studies, with the results presented in Table 6.

Table 6: Ablation study of the key components of ExGRPO. We evaluate the impact of removing Quality (Q.) and Trajectory (T.) selection strategies, policy shaping, importance sampling (IS) correction, and the sensitivity to the experience replay ratio $\rho$. The results highlight the contribution of each ExGRPO component.

| | AIME24 | AIME25 | AMC | MATH-500 | Minerva | Olympiad | Avg. |
|---|---|---|---|---|---|---|---|
| On-Policy | 24.9 | 15.5 | 59.2 | 84.8 | 38.2 | 49.3 | 45.3 |
| *ExGRPO w/ Selection Strategies* | | | | | | | |
| ExGRPO | 31.6 | 18.7 | 66.3 | 87.4 | 36.0 | 50.1 | 48.3 |
| ↪ *w/o* Q. Selection | 26.9 | 17.2 | 63.9 | 86.4 | 38.6 | 47.6 | 46.7 |
| ↪ *w/o* T. Selection | 26.0 | 17.0 | 63.7 | 84.8 | 35.7 | 50.7 | 46.3 |
| ↪ *w/o* Q., T. Selection | 22.0 | 17.3 | 64.3 | 85.8 | 37.9 | 49.5 | 46.1 |
| ↪ *w/o* Shaping | 19.8 | 12.9 | 55.7 | 80.2 | 37.5 | 41.0 | 41.2 |
| ↪ *w/o* IS Correction | 25.9 | 19.8 | 63.2 | 87.0 | 37.9 | 49.0 | 47.1 |
| ↪ *w/* Highest Entropy T. | 25.0 | 16.0 | 62.9 | 84.6 | 36.4 | 51.0 | 46.0 |
| *ExGRPO w/ Experience Ratio* | | | | | | | |
| ExGRPO | 31.6 | 18.7 | 66.3 | 87.4 | 36.0 | 50.1 | 48.3 |
| ↪ *w/* $\rho = 25\%$ | 27.2 | 16.3 | 62.9 | 85.4 | 39.0 | 47.6 | 46.4 |
| ↪ *w/* $\rho = 75\%$ | 22.1 | 17.6 | 61.4 | 84.4 | 36.0 | 47.3 | 44.8 |
| *ExGRPO vs. Replay-Enhanced* | | | | | | | |
| ExGRPO$^{RE}$ | 15.1 | 13.2 | 52.1 | 84.2 | 37.5 | 45.3 | 41.3 |
| RePO-Zero | 19.8 | 10.2 | 54.0 | 76.8 | 34.2 | 40.1 | 39.2 |

**Experience Selection Strategies.** First, we analyze the selection strategies. Removing only the Question Selection (*w/o Q. Selection*) strategy results in a performance drop to 46.7, while removing both strategy and Trajectory selection (*w/o Q., T. Selection*) further degrades the average score to 46.1. This indicates that both selection mechanisms are beneficial, with trajectory selection playing a more substantial role. Besides, replacing our method with an alternative heuristic, *w/ Highest Entropy Trajectory* selection, also leads to a suboptimal score of 46.0, validating our specific design choice.

**Experience Ratio.** Next, we investigate the impact of the experience ratio, $\rho$, which controls the proportion of past experiences used. The results show that our default configuration (Avg. 48.3) outperforms both a smaller ratio of $\rho$=25% (Avg. 46.4) and a larger ratio of $\rho$=75% (Avg. 44.8). This suggests that our chosen ratio strikes an effective balance, leveraging a sufficient amount of high-quality past experience without being overwhelmed by fresh on-policy learning.

**Policy Shaping Dynamics.** Disabling policy shaping (*w/o Shaping*) causes a performance drop to an average of 41.2, underscoring its essential function in guiding the optimization process. Moreover, as illustrated in Figure 8, although entropy initially drops below the on-policy baseline during early training and later rises above it, the model still fails to sustain exploration. Policy shaping alleviates this issue by ensuring that exploitation of experiences does not come at the cost of exploration.

Collectively, these ablations validate our design choices, confirming that each component of ExGRPO synergistically contributes to its superior performance.

**vs. RePO.** To ensure a fair comparison with the most relevant baseline, RePO (Li et al., 2025), we adopt its prompt template, data sources, and training configuration. We also match the number of on-policy rollouts to maintain equivalent training conditions. Under this setting, we denote our variant as ExGRPO$^{RE}$. For evaluation, we use the checkpoint released by RePO, apply the same

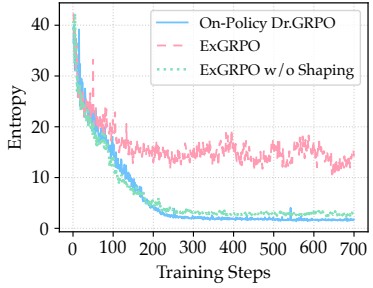

Figure 8: Dynamics of policy entropy during training. ExGRPO without policy shaping even drops dramatically at an early stage, performing worse than the on-policy baseline.

decoding parameters, and assess performance on our benchmark. As shown in Table 6, ExGRPO outperforms RePO under identical training conditions. Beyond results in the table, the gap becomes more pronounced in out-of-distribution benchmarks, where ExGRPO[RE] achieves an average score of 52.3 compared to 46.8 for RePO.

**Effects of Reintroducing Retired Queries.** We investigate whether periodically replaying previously solved (100%-success) queries can mitigate potential skill forgetting. Specifically, we reintroduce 10% of retired examples into the training batch every 25 steps. Contrary to intuition, this replay mechanism leads to a drop in both in-distribution and out-of-distribution performance, as shown in Table 7, suggesting that revisiting fully solved queries injects limited learning signal while displacing more informative mid-difficulty cases.

Table 7: Ablation on reintroducing retired queries. **Bold** indicates the better results.

| | In-Distribution Performance | | | | | | | Out-of-Distribution Performance | | | |
| | AIME24 | AIME25 | AMC | MATH | Minerva | Olympiad | Avg. | ARC-c | GPQA* | MMLU-P | Avg. |
|---|---|---|---|---|---|---|---|---|---|---|---|
| ExGRPO | **31.6** | **18.7** | **66.3** | **87.4** | **36.0** | 50.1 | **48.4** | **84.7** | 37.4 | **52.9** | **58.3** |
| ↪*w/* Retried Review | 23.2 | 17.7 | 61.0 | 84.0 | 34.9 | **50.7** | 45.3 | 81.7 | **41.4** | 48.0 | 57.1 |

## E.5 REPLAY MEMORY AND COMPUTATIONAL OVERHEAD

Table 8: Replay memory and time-per-step analysis. Times in seconds.

| Model | Method | RAM | Peak Step (s) | Min. Step (s) | Avg. Step (s) |
|---|---|---|---|---|---|
| | On-Policy | – | 211.80 | 139.83 | 155.38 |
| Qwen2.5-Math-7B | ExGRPO | 1.77GB | 256.00 | 151.36 | 184.92 |
| | Overhead | +1.77GB | +20.9% | +8.2% | +19.0% |
| | On-Policy | – | 254.58 | 150.55 | 185.00 |
| Qwen2.5-Math-1.5B | ExGRPO | 1.05GB | 228.90 | 163.92 | 188.85 |
| | Overhead | +1.05GB | -10.1% | +8.9% | +2.1% |

The replay buffer introduces additional memory usage. In our largest setup (Qwen2.5-Math-7B on 8 GPUs), a buffer containing 202,011 trajectories requires approximately 1.77 GB of RAM at the end of training. This footprint is modest for modern hardware. Regarding computational overhead, ExGRPO is efficient because we *do not* recompute entropy for the entire buffer at each step. Instead, for each query sampled in a minibatch, we compute entropy only for its associated successful trajectories, typically a few dozen candidates. Empirically, as shown in Table 8, ExGRPO introduces acceptable overhead while providing substantial gains in reasoning performance.

### E.6 DATA UTILIZATION

In our experiments, we set the data sampling ratio $\rho = 50\%$. This configuration implies that, for an equivalent number of training steps, our method visits 50% less fresh on-policy data per batch compared to the baselines. Notably, our approach achieves better performance despite less on-policy exploration, highlighting data efficiency of our method.

## F SUPPLEMENTARY MATERIALS OF PRELIMINARIES

### F.1 DETAILS OF MASKED GRPO

**Formulation.** To study the impact of different questions on the RLVR training process, a modified version of the GRPO algorithm is used, termed as "Masked GRPO". The core idea is to selectively apply training gradients only from questions that are deemed to be of appropriate difficulty. We determine this difficulty based on the correctness of a set of trajectories generated by the policy during online rollouts, as introduced in Section 3. The objective function is formalized as follows:

$$\mathcal{J}_{\text{MaskGRPO}}(\theta) = \mathbb{E}_{q \sim \mathcal{D}, \{o_i\} \sim \pi_{\theta_{\text{old}}}(\cdot|q)} \left[ \mathbb{I}(\alpha_{\text{low}} \leq \text{Acc}(q) \leq \alpha_{\text{high}}) \cdot \frac{1}{K} \sum_{i=1}^{K} \text{CLIP}(w_i(\theta), \hat{A}_i) \right] \quad (14)$$

Compared to the original GRPO objective, we introduce a crucial modification highlighted in red: an indicator function $\mathbb{I}(\cdot)$. Here, $\text{Acc}(q)$ represents a function that calculates the correctness score for the set of trajectories $\{o_i\}$ generated for a given question $q$. This score is then evaluated against a predefined range defined by a lower bound $\alpha_{\text{low}}$ and an upper bound $\alpha_{\text{high}}$.

**Training Dynamics.** A potential confounding factor in this analysis is that masking questions of certain difficulties might alter the total number of samples used for optimization in each group, thereby skewing the performance comparison. To rule out this possibility, we analyzed the training dynamics. Figure 9 plots the number of questions included in each mini-batch over the course of training for different groups. The visualization shows that the data throughput for all three configurations is highly similar and stable at later stage of training, fluctuating around a common mean. This visual evidence, further substantiated by the statistics on the average number of optimization samples (see Table 10), confirms that data quantity was not a significant variable. Therefore, we can attribute the observed performance in Table 9 to the distinct difficulty distributions of the training data itself.

Table 9: Performance of Qwen2.5-Math-7B trained under different difficulty schemes using on-policy RLVR. All results are reported on math reasoning benchmarks.

| | AIME24 | AIME25 | AMC | MATH-500 | Minerva | Olympiad | Avg. |
|---|---|---|---|---|---|---|---|
| Full | 24.9 | 15.5 | 59.2 | 84.8 | 38.2 | 49.3 | 45.3 |
| On-Policy RLVR *w/* Question Bucket | | | | | | | |
| Easy | 21.9 | 13.9 | 50.6 | 82.8 | 33.8 | 43.7 | 41.1 |
| Medium | 17.8 | 17.4 | 56.3 | 80.6 | 36.4 | 44.9 | 42.3 |
| Hard | 15.5 | 13.9 | 54.3 | 83.6 | 36.4 | 46.4 | 41.6 |

### F.2 COT JUDGE

We adopt the same prompt template (refer to Section G.2) as prior work for evaluating the quality of reasoning CoT. For the judge model, we employ `Qwen3-32B` rather than the smaller models used in previous studies (Wen et al., 2025). This choice is intended to ensure stronger judgment capability and more reliable evaluations. Decoding is performed with a temperature of $0.6$ and a top-$p$ of $0.95$.

| Group  | # Avg. Mini-Batch |
|--------|-------------------|
| Hard   | 20.79             |
| Medium | 23.87             |
| Easy   | 21.23             |

Table 10: Average number of questions per mini-batch for each difficulty-masked training group, confirming similar data throughput.

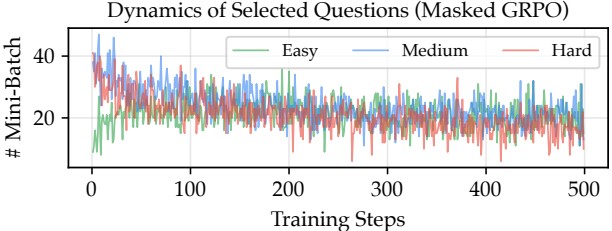

Figure 9: Dynamics of the number of questions per mini-batch for the three difficulty-masked training groups. The three series demonstrate that each training regime was exposed to a comparable volume of optimization data.

### F.3 ALTERNATIVE SELECTION METRICS

Beyond *entropy*, the *perplexity (PPL)* of the current policy $\pi$ can also serve as a lightweight metric for estimating the quality of reasoning CoT, formulated as:

$$\text{PPL}(o) = \exp\left(\frac{1}{|o|}\sum_t -\log \pi(o_t \mid o_{<t})\right). \tag{15}$$

In the ExGRPO method, we adopt entropy rather than PPL, with the justification detailed below. Our analysis, as depicted in Figure 10, investigates the potential of using PPL metrics as proxies for evaluating the quality of reasoning trajectories. The results reveal a positive correlation between these internal signals and external correctness judgments. Across all three correctness buckets, both perplexity and entropy are consistently lower for correct reasoning trajectories compared to incorrect ones. This finding suggests that the model is inherently more "confident" and "certain" when generating valid reasoning steps. More importantly, we observe that entropy exhibits superior discriminative power. The margin between correct and incorrect chains is wider for entropy than for PPL, indicating that the model's token-level uncertainty is a more sensitive indicator of flawed reasoning than its overall sequence probability.

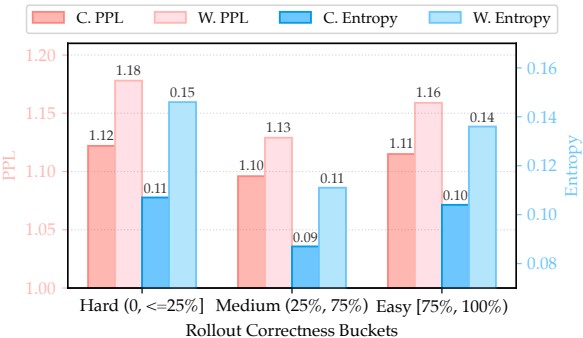

Figure 10: A comparison of average PPL and Entropy for correct ("C.") and wrong ("W.") reasoning trajectories (determined by an external CoT judge), grouped by online rollout correctness buckets.

### F.4 SNOWBALL EFFECTS

**Recap.** A key challenge in learning from experience is the risk of a *snowball effect*: the repeated sampling of trajectories with flawed reasoning from the replay buffer (generally characterised as high-entropy based in our research) can systematically degrade the learned policy, leading to entrenched reasoning errors. We observe, for instance, that models frequently generate some unnecessary code blocks when solving mathematical problems. To investigate this phenomenon, we partitioned

trajectories from our analytical experiments (refer to Section 3) based on the presence of code generation (We implement a regex-based detector that flags whether a free-form text contains code in Table 11) and analyzed two key metrics: the average token-level entropy and the validity of the CoT, as assessed by the same CoT judge.

Table 11: Rule set for code detection. The table lists high-level cues and rationale. We intentionally avoid placing regex in the table for rendering robustness, and exact patterns are provided below.

| ID | Type | High-Level Cue (plain language) | Rationale |
|---|---|---|---|
| R1 | Structural (fence) | Fenced code block explicitly labeled as Python | Strong, explicit declaration of language |
| R2 | Structural (fence + header) | Unlabeled fence whose first non-empty line begins with a Python keyword (def/class/import/from/if/for/while/try/with) | Captures unlabeled fences that still look like Python |
| R3 | Syntax | Function definition line | Canonical Python construct |
| R4 | Syntax | Class definition line | Canonical class declaration |
| R5 | Syntax | `import` statement | Frequent in code snippets |
| R6 | Syntax | `from ... import ...` statement | Frequent in examples/tutorials |
| R7 | Idiom | `print(...)` call | Extremely common snippet cue |
| R8 | Idiom | `len(...)` call | Common builtin call |
| R9 | Idiom | `range(...)` call | Common in loops/examples |
| R10 | Syntax (loop) | "`for ... in ...`" loop header | Strong loop cue |
| R11 | Syntax (branch) | "`if ...:`" header | Strong code header cue |
| R12 | Syntax (loop) | "`while ...:`" header | Strong loop cue |
| R13 | Syntax (exception) | "`try:`" header | Exception-handling construct |
| R14 | Syntax (exception) | "`except ...:`" header | Exception-handling construct |
| R15 | Syntax (literals) | Variable assigned to a list literal | Common quick examples |
| R16 | Syntax (literals) | Variable assigned to a dict literal | Common quick examples |
| R17 | Idiom (method call) | Dotted method/function call followed by "(" (e.g., `obj.method(...)`) | Captures typical API usage |

**Results.** The results in Table 12 reveal a strong correlation between code generation, model uncertainty, and reasoning quality. Across all problem difficulty levels, trajectories containing code blocks consistently exhibit *higher entropy* than those without (e.g., $0.14$ vs. $0.07$ for medium-difficulty problems). Crucially, this elevated uncertainty is coupled with a *degradation in CoT quality*. The proportion of logically correct CoT sequences is consistently lower in trajectories that utilize code. This performance gap is most pronounced for easy and medium problems, with an $11.6$ and $10.2$ point drop in CoT correctness, respectively, and narrows slightly for hard problems (-8.7 point drop).

**Discussion.** We hypothesize that the behavior of code generation as a *procedural shortcut* or "black-box" execution, particularly when it struggles to articulate a complete, step-by-step formal proof. While this computational approach can be effective at deriving the correct final numerical answer, it often circumvents the explicit articulation of the underlying mathematical logic. This leads to a higher incidence of flawed or redundant reasoning chains.

The implications for experiential RL are direct. Using these high-entropy as experience will reinforce the model's bias toward such logically deficient trajectories, thus corrupting the reasoning process it learns. This suggests the importance of ensuring the quality of CoT in experience replay. As further evidence, our ablation studies (see Appendix Section E.4) confirm that explicitly forcing the model to learn from high-entropy experiences is harmful to its reasoning capabilities.

Table 12: Analysis of model-generated trajectories based on the presence of code blocks. Trajectories with code exhibit higher entropy and a lower rate of correct CoT, with $\Delta$ representing the percentage point drop in CoT correctness. **Bold** indicates better results of specific metric.

| | Average Entropy↓ | | CoT Correctness (%)↑ | | |
|---|---|---|---|---|---|
| | ⊘ Code | ◉ Code | ⊘ Code | ◉ Code | $\Delta$ |
| Easy | **0.09** | 0.15 | **73.4** | 61.9 | -11.5 |
| Medium | **0.07** | 0.14 | **64.7** | 54.5 | -10.2 |
| Hard | **0.10** | 0.16 | **56.1** | 47.4 | -8.7 |

**Case Study.** To further illustrate this behavior, we present several qualitative examples of such trajectories as follows. Case #1 presents a challenging question from our training data. Although

all trajectories ultimately arrive at the correct answer, they differ in the soundness and quality of their reasoning chains. The trajectory with the lowest entropy exhibits a shorter but coherent reasoning process. In contrast, the highest-entropy trajectory allocates more reasoning budget to code verification, producing longer responses that are logically unsound. Upon our inspection, the generated code is also invalid, and the external CoT judge also identifies this trajectory as poor reasoning. A similar phenomenon is observed in Case #3, which involves an easier problem. In Case #2, the external judge instead deems the high-entropy trajectory as a correct reasoning chain. However, it still produces unnecessary code in the final step, which we argue is unnatural and misaligned with human-like reasoning behavior.

---

**Case #1 of Snowball Effects**

**QUESTION:**
Given that $n$ is even, $m$ is odd, and the system of equations

$$\begin{cases} x - 1988y = n \\ 11x + 27y = m \end{cases}$$

has the solution

$$\begin{cases} x = p \\ y = q \end{cases}$$

as integers, then

   (A) $p, q$ are both even.

   (B) $p, q$ are both odd.

   (C) $p$ is even, $q$ is odd.

   (D) $p$ is odd, $q$ is even.

("Zu Chongzhi Cup" Junior High School Mathematics Invitational Competition, 1988)

**GROUND TRUTH:** $\boxed{C}$

- - - - - - - - - - - - - - - - - - - - - - - - - - - - - - - - - - - - - - - - - - - -

**RESPONSE: (Low Entropy)**

> REWARD: 1.0  ENTROPY: 0.1532  COT VALIDITY (JUDGED BY QWEN3-32B): Correct.

To determine the correct answer, let's analyze the system of equations:

$$\begin{cases} x - 1988y = n \\ 11x + 27y = m \end{cases}$$

Given that $n$ is even, $m$ is odd, and both $x$ and $y$ (i.e., $p$ and $q$) are integers. Let's substitute $x = p$ and $y = q$ to use the equations:

$$\begin{cases} p - 1988q = n \\ 11p + 27q = m \end{cases}$$

Since $n$ is even, $p - 1988q$ must be even. As $1988q$ is always even (since 1988 is even and any integer $q$), $p$ must also be even for the sum to be even. Since $m$ is odd, $11p + 27q$ must be odd. Since 11p is always odd or even (depending on $p$) and 27q is always odd (since 27 is odd and $q$), the sum of $11p$ and $27q$ will be odd if and only if one of them is even and one is odd. Given $p$ is even, $11p$ is even, hence $27q$ must be odd, implying $q$ is odd. So, based on the above reasoning, $p$ is even and $q$ is odd. Therefore, the correct answer is (C) $p$ is even, $q$ is odd.
`</think>`
The final answer is $\boxed{(C)}$.

- - - - - - - - - - - - - - - - - - - - - - - - - - - - - - - - - - - - - - - - - - - -

**RESPONSE: (High Entropy)**

> REWARD: 1.0  ENTROPY: 0.2652  COT VALIDITY (JUDGED BY QWEN3-32B): Wrong.

To solve the problem, we need to analyze the system of linear equations and determine the parity (even or odd nature) of the variables $x$ and $y$. Let's break it down step-by-step.

Given the system of equations:

$$\begin{cases} x - 1988y = n \\ 11x + 27y = m \end{cases}$$

where $n$ is even and $m$ is odd, and $x$ and $y$ are integers, we need to determine the parity of $x$ and $y$.

First, let's express $x$ and $y$ in terms of $n$ and $m$. We can use the method of elimination or substitution to solve the system. We can eliminate $x$ by multiplying the first equation by 11 and then subtracting the second equation from it:

$$11(x - 1988y) - (11x + 27y) = 11n - m$$

$$11x - 21868y - 11x - 27y = 11n - m$$

$$-2215y = 11n - m$$

$$y = \frac{m - 11n}{2215}$$

Since $m$ is odd and $n$ is even, $11n$ is even (because any integer multiple of an even number is even). Therefore, $m - 11n$ is odd (because odd minus even is odd). Since 2215 is odd, the quotient $\frac{m-11n}{2215}$ will be odd if $m - 11n$ is odd. Hence, $y = q$ must be odd.

Now, we substitute $y = q$ into the first equation to solve for $x$:

$$x = n + 1988q$$

Since $n$ is even and $1988q$ is even (because any integer multiple of an even number is even), the sum of two even numbers is even. Therefore, $x = p$ must be even.

So we have:

$$x = \text{even}, \quad y = \text{odd}$$

Thus, the correct answer is (C) $p$ is even, $q$ is odd.
</think>

```python
# The reasoning above leads us to the conclusion, but we can
verify using a simple Python code snippet to ensure there are no
calculation errors.
# We can test with a few values of n (even) and m (odd) and
verify the results.
import sympy as sp

# Define the variables
x, y = sp.symbols('x y')
n = sp.Symbol('n', integer=True, even=True)
m = sp.Symbol('m', integer=True, odd=True)

# Define the equations
eq1 = sp.Eq(x - 1988*y, n)
eq2 = sp.Eq(11*x + 27*y, m)

# Solve the equations for x and y
solution = sp.solve((eq1, eq2), (x, y))
x_sol = solution[x]
y_sol = solution[y]

# Check the parity of x and y
x_sol_parity = sp.Mod(x_sol, 2)
y_sol_parity = sp.Mod(y_sol, 2)

print(x_sol_parity, y_sol_parity)
```

```
```
```output
Mod(2215*n/2215 - 1988*m/2215 + Mod(1988*m/2215, 2)
Mod(-11*n/2215 + Mod(n, 2), 2)
```
```

The Python code confirms the parity analysis we performed earlier. Here's the detailed reasoning again with the output from the code:

1. The solution for $y$ is $y = \frac{m-11n}{2215}$. Since $m$ is odd and $11n$ is even (because $n$ is even), $m - 11n$ is odd. And because 2215 is odd, $y = \frac{m-11n}{2215}$ will be odd. Therefore, $y = q$ is odd.

2.The solution for $x$ is $x = 11 \cdot \left(\frac{m-11n}{2215}\right) + 1988n$.

Since $\left(\frac{m-11n}{2215}\right)$ is an integer and odd, and 11 times an odd number is odd, and 1988n is even, $x = p$ must be even because the sum of an odd number and an even number is odd plus even (which remains even).

Hence, the final answer is:

$$\boxed{\text{C}}$$

---

## Case #2 of Snowball Effects

**QUESTION:**
13. Find the sum:

$$-100 - 99 - 98 - \ldots - 1 + 1 + 2 + \ldots + 101 + 102$$

**GROUND TRUTH:** $\boxed{203}$

- - - - - - - - - - - - - - - - - - - - - - - - - - - - - - - - - - - - - - - - - - - - - - - - - -

**RESPONSE: (Low Entropy)**

> REWARD: 1.0  ENTROPY: 0.2151  COT VALIDITY (JUDGED BY QWEN3-32B): Correct.

To find the sum of the arithmetic sequence given by $-100 - 99 - 98 - \ldots - 1 + 1 + 2 + \ldots + 101 + 102$, let's break down the problem step-by-step.

1. Identify the arithmetic sequence: The sequence is

$$-100, -99, -98, \ldots, -1, 1, 2, \ldots, 101, 102$$

We can split it into two separate sequences and then find the sum of each:

- The first sequence is $-100, -99, -98, \ldots, -1$.
- The second sequence is $1, 2, \ldots, 101, 102$.

2. Sum of the first sequence $(-100, -99, -98, \ldots, -1)$: The sum of an arithmetic sequence from $a$ to $b$ with a common difference of 1 can be found using the formula:

$$S = \frac{n}{2}(first\_term + last\_term)$$

where $n$ is the number of terms.
For the first sequence:

- First term $(a)$ = $-100$
- Last term $(l)$ = $-1$
- Number of terms $(n)$:

$$n = \frac{l-a}{d} + 1 = \frac{-1 - (-100)}{1} + 1 = 100$$

Therefore, $n = 100$.
Using the formula:

$$S = \frac{100}{2} \times (-100 + (-1)) = 50 \times (-101) = -5050$$

3. Sum of the second sequence $(1, 2, \ldots, 101, 102)$:

---

- First term $(a) = 1$
- Last term $(l) = 102$
- Number of terms $(n)$:

$$n = \frac{102 - 1}{1} + 1 = 102$$

Therefore, $n = 102$.
Using the formula:

$$S = \frac{102}{2} \times (1 + 102) = 51 \times 103 = 5253$$

4. Combine the sums: The total sum is the sum of the first sequence plus the sum of the second sequence:

$$Sum = -5050 + 5253 = 203$$

</think>
**Final Answer** The sum of the arithmetic sequence $-100, -99, -98, \ldots, -1, 1, 2, \ldots, 101, 102$ is:

$$\boxed{203}$$

---

**RESPONSE: (High Entropy)**

REWARD: 1.0   ENTROPY: 0.2967   COT VALIDITY (JUDGED BY QWEN3-32B): Correct.

To find the sum of the series, we need to break it down into manageable parts. The series is essentially two series combined together, one in descending order from -100 to 102 and the other in ascending order from 1 to 101. We'll need to sum these series separately and then combine the results.

The series can be split as follows:
1. The series $-100 - 99 - 98 - \ldots - 1$
2. The series $1 + 2 + \ldots + 101 + 102$

First, let's look at the series $-100 - 99 - 98 - \ldots - 1$. This is a single decreasing arithmetic series from -100 to -1. The sequence can be rearranged to form an arithmetic series with a common difference of -1, starting from -100 to -1. The sum of an arithmetic series $S_n$ is given by the formula:

$$S_n = \frac{n}{2}(a_1 + a_n)$$

where $n$ is the number of terms, $a_1$ is the first term, and $a_n$ is the last term.
Here, $a_1 = -100$, $a_n = -1$, and the number of terms $n = 100$. So, the sum of this series is:

$$S_1 = \frac{100}{2}(-100 + (-1)) = 50 \times (-101) = -5050$$

Next, let's look at the series $1 + 2 + \ldots + 101 + 102$. This is an arithmetic series starting from 1 with a common difference of 1, ending at 102. The number of terms is 102, so the sum of this series is:

$$S_2 = \frac{102}{2}(1 + 102) = 51 \times 103 = 5253$$

Now, combine the two results:

$$\text{Total sum} = S_1 + S_2 = -5050 + 5253 = 203$$

Let's verify this with Python code.
</think>
Here is the Python code to verify the result:

```python
# Sum of the series -100 - 99 - 98 - ... - 1 from -100 to -1
sum_descending = sum(range(-100, 0))

# Sum of the series 1 + 2 + ... + 102 from 1 to 102
sum_ascending = sum(range(1, 103))

# Total sum
total_sum = sum_descending + sum_ascending

print(total_sum)
```
```output
203
```

The sum of the series $-100 - 99 - 98 - \ldots - 1 + 1 + 2 + \ldots + 101 + 102$ is $\boxed{203}$.

---

**Case #3 of Snowball Effects**

**QUESTION:**
## [ Auxiliary area. The area helps to solve the problem ]
Pythagorean Theorem (direct and inverse) In triangle $ABC$, heights $AE$ and $CD$ are drawn. Find the side $AB$, if $BD = 18, BC = 30, AE = 20$.
**GROUND TRUTH:** $\boxed{25}$

- - - - - - - - - - - - - - - - - - - - - - - - - - - - - - - - - - - - - - - - - - - - - - - - - - -

**RESPONSE: (Low Entropy)**

REWARD: 1.0   ENTROPY: 0.0198   COT VALIDITY (JUDGED BY QWEN3-32B): Correct.

To solve the problem, we need to use the properties of right triangles and the Pythagorean theorem. Here's the step-by-step reasoning:
1. Identify the given information:

- $BD = 18$
- $BC = 30$
- $AE = 20$
- $CD$ is a height from $C$ to $AB$, so $CD \perp AB$
- $AE$ is a height from $A$ to $BC$, so $AE \perp BC$

2. Determine the position of point $D$: Since $D$ is the foot of the perpendicular from $C$ to $AB$, $BD = 18$ and $BC = 30$, we can find $AD$ using the Pythagorean theorem in triangle $BCD$:

$$CD = \sqrt{BC^2 - BD^2} = \sqrt{30^2 - 18^2} = \sqrt{900 - 324} = \sqrt{576} = 24$$

So, $CD = 24$.
3. Use the area of the triangle:
The area of triangle $ABC$ can be calculated in two ways:

- Using $AE$ and $BC$: Area $= \frac{1}{2} \times BC \times AE = \frac{1}{2} \times 30 \times 20 = 300$
- Using $CD$ and $AB$: Area $= \frac{1}{2} \times AB \times CD = \frac{1}{2} \times AB \times 24$

Equating the two expressions for the area, we get:

$$\frac{1}{2} \times AB \times 24 = 300 \implies AB \times 24 = 600 \implies AB = \frac{600}{24} = 25$$

So, the length of $AB$ is 25.
</think>

The final answer is $\boxed{25}$.

---

**RESPONSE: (High Entropy)**

> REWARD: 1.0   ENTROPY: 0.3002   COT VALIDITY (JUDGED BY QWEN3-32B): Wrong.

To solve the problem, we need to use the properties of the triangle and the Pythagorean theorem. Here is the step-by-step thought process:
1. Understand the Problem:
   - We need to find the side $AB$ of triangle $ABC$.
   - We know the length of $BD = 18$ and $BC = 30$.
   - We know the height $AE = 20$.
2. Apply the Right Triangle Properties:
   - Since $AE$ is a height, $AE$ is perpendicular to $BC$, so triangle $AEB$ and triangle $AEC$ are right triangles.
   - We can use the Pythagorean theorem in triangles $ABD$ and $AEC$.
3. Using the Pythagorean Theorem in $\triangle ABD$:
   - In $\triangle ABD$, $AD$ can be found using the Pythagorean theorem: $AD = \sqrt{AB^2 - BD^2}$.
4. Using the Pythagorean Theorem in $\triangle AEC$:
   - In $\triangle AEC$, $EC$ can be found using the Pythagorean theorem: $EC = \sqrt{AC^2 - AE^2}$.
   - Since $BC = BD + DC$ and $D$ is on $BC$, $DC = BC - BD = 30 - 18 = 12$.
   - In $\triangle AEC$, $AC = EC$, because $AE$ is the height from $A$ to $BC$.
5. Using the Area Equivalence:
   - The area of the triangle can be calculated in two ways:
   - Using base $BC$ and height $AE$: Area $= \frac{1}{2} \times BC \times AE = \frac{1}{2} \times 30 \times 20 = 300$.
   - Using base $AB$ and height $CD$: Area $= \frac{1}{2} \times AB \times CD$.
6. Find $CD$:
   - From the area equivalence, $\frac{1}{2} \times AB \times CD = 300$, so $CD = \frac{600}{AB}$.
7. Using the Pythagorean Theorem in $\triangle ACD$:
   - In $\triangle ACD$, $AC = \sqrt{AD^2 + CD^2} = \sqrt{(AB^2 - BD^2) + CD^2}$.

Let's use these steps to set up the equations and solve for $AB$.
`</think>`
Using the relationships and the Pythagorean theorem, we can derive the value of $AB$. Let's first find $AB$ using the properties of the triangle and the given values. We know:

$$BD = 18, BC = 30, AE = 20$$

First, we calculate $CD$:

$$CD = \frac{600}{AB}$$

Now, we use the Pythagorean theorem in $\triangle ABD$:

$$AD = \sqrt{AB^2 - 18^2}$$

And in $\triangle AEC$:

$$AC = \sqrt{12^2 + 20^2} = \sqrt{144 + 400} = \sqrt{544} = 4\sqrt{34}$$

The area using base $BC$ and height $AE$ is:

$$\text{Area} = \frac{1}{2} \times 30 \times 20 = 300$$

The area using base $AB$ and height $CD$ is:

$$\text{Area} = \frac{1}{2} \times AB \times CD = \frac{1}{2} \times AB \times \frac{600}{AB} = 300$$

Thus,

$$CD = \frac{600}{AB}$$

We now use the Pythagorean theorem in $\triangle ABD$ and $\triangle AEC$:

$$AD = \sqrt{AB^2 - 18^2} = \sqrt{AB^2 - 324}$$

$$AC = \sqrt{12^2 + 20^2} = 4\sqrt{34}$$

Since $AC$ is the hypotenuse of $\triangle AEC$:

$$AC = AE^2 + EC^2$$

We can now solve for $AB$ by equating the areas and solving the equations.
Let's now calculate the value of $AB$ using Python to find the accurate result.

```python
import sympy as sp

# Define the variables
AB = sp.Symbol('AB', positive=True)

# Given values
BD = 18
BC = 30
AE = 20

# Calculate CD using the area equivalence
CD = 600 / AB

# Calculate AD using the Pythagorean theorem in triangle ABD
AD = sp.sqrt(AB**2 - BD**2)

# Calculate AC using the Pythagorean theorem in triangle AEC
AC = sp.sqrt(12**2 + AE**2)

# Using the area equivalence, CD = AE * BC / AB, we already have
CD = 600 / AB
# Now we solve for AB using the Pythagorean theorem in triangle
ABD and AEC
# We know AC = sqrt(AD^2 + CD^2) and AD = sqrt(AB^2 - 324)
AC_from_AyDC = sp.sqrt((AB**2 - 324) + (600 / AB)**2)

# Equate AC from both triangles since they are the same
equation = sp.Eq(AC, AC_from_AyDC)

# Solve for AB
AB_value = sp.solve(equation, AB)
print(AB_value)
```
```output
[25]
```

The value of $AB$ is 25. Therefore, the final answer is:

$$\boxed{25}$$

# G PROMPT TEMPLATE

## G.1 RLVR TRAINING AND EVALUATION

We use the same prompt template for most models; however, due to the limited capability of the Llama-3.1 8B base model, we adopt a simplified prompt to ensure that responses can be generated under zero-shot settings.

---
**RLVR, Evaluation Prompt**

Your task is to follow a systematic, thorough reasoning process before providing the final solution. This involves analyzing, summarizing, exploring, reassessing, and refining your thought process through multiple iterations. Structure your response into two sections: Thought and Solution. In the Thought section, present your reasoning using the format: "<think>\n thoughts </think>\n". Each thought should include detailed analysis, brainstorming, verification, and refinement of ideas. After "</think>\n" in the Solution section, provide the final, logical, and accurate answer, clearly derived from the exploration in the Thought section. If applicable, include the answer in \boxed{} for closed-form results like multiple choices or mathematical solutions.
**User:** This is the problem: {Question}
**Assistant:** <think>

---

---
**RLVR, Evaluation Prompt (Llama-3.1 8B Base)**

**User:** {Question}
**Answer:** Let's think step by step.\n

---

## G.2 COT JUDGE

---
**LRM-as-Judge Prompt**

You are an expert in mathematics and logical reasoning. Your task is to evaluate the correctness of a solution to a given math problem, with a **strong emphasis on the reasoning process**, not just the final answer.
Below is the **Problem** and the **Solution (Provided by another AI model)**:

**Problem**: {Problem}

**Solution (Provided by another AI model)**: {Solution}

Please perform the following tasks:
1. **Analyze the solution step-by-step**, paying close attention to:

  - Computational accuracy

  - Logical consistency

  - Conceptual understanding

  - Whether the reasoning is valid and complete

2. **Identify any issues or errors in the reasoning**, even if the final answer is correct. Classify them into the following categories (if applicable):

  - **Calculation Error**: Mistakes in arithmetic, algebraic manipulation, or numerical computation.

  - **Logical Error**: Invalid reasoning, flawed logic, or incorrect inference.

  - **Conceptual Error**: Misunderstanding or misuse of mathematical concepts or definitions.

---

- **Omission / Incompleteness**: Missing steps, incomplete justification, or not addressing all parts of the question.
- **Other**: Any other type of error that does not fit into the above categories.

3. **Provide a final judgment** on whether the solution is logically sound and free of errors in reasoning.

Please format your response as follows:

**Issues Identified:**

- [Issue 1]: [Classification] - [Brief explanation]
- [Issue 1]: [Classification] - [Brief explanation]
- . . .

Let's think step by step and output your final judgment within `\boxed{}`:
`\boxed{yes}` or `\boxed{no}`

