# OpenReview forum: "ExGRPO: Learning to Reason from Experience"
_ICLR.cc/2026/Conference — ICLR 2026 Poster_

### Official Review · Reviewer_wSTN · 2025-10-14

**Soundness:** 3
**Presentation:** 4
**Contribution:** 2
**Rating:** 8
**Confidence:** 3

**Summary:**

This paper proposes ExGRPO (Experiential Group Relative Policy Optimization), a reinforcement learning method tailored for large language models (LLMs) in the RLVR (Reinforcement Learning from Verifiable Rewards) paradigm. ExGRPO addresses the inefficiencies of on-policy RLVR, which discards prior successful trajectories, by designing a principled experience management pipeline. The key innovations include:

Identifying rollout correctness and trajectory entropy as indicators of experience quality;

Organizing trajectories by correctness buckets and selecting low-entropy paths for replay;

A mixed-policy optimization objective that blends off-policy experience replay with on-policy rollouts, using importance reweighting and policy shaping to control exploration–exploitation tradeoffs;

A delayed start mechanism to avoid low-quality replay at early stages.

Extensive experiments on multiple reasoning benchmarks (AIME, MATH-500, Olympiad, ARC-c, GPQA, etc.) and across backbone models (Qwen, LLaMA, LUFFY, etc.) demonstrate consistent performance gains (+3.5/+7.6 points in-domain/OOD) and improved stability over strong baselines like GRPO and RePO.

**Strengths:**

Well-motivated problem: The paper targets a real bottleneck in RLVR—wasteful discard of prior trajectories—and connects this with known issues in experience replay for RL and LLMs.

Principled design of experience value: The use of rollout correctness buckets and entropy as quality proxy is backed by a detailed analysis (Fig.1), offering clear guidance for trajectory selection.

Strong methodology: ExGRPO is conceptually elegant and practically implementable, combining well-established principles (importance weighting, entropy shaping) into a novel RLVR-specific pipeline.

Thorough evaluation:

Tested on five models (1.5B–8B);

Covers both in-distribution (math) and out-of-distribution reasoning tasks;

Includes ablation studies (Fig.7), training stability analysis (Fig.4), and replay dynamics (Fig.5–6);

Achieves SOTA or near-SOTA results (Table 1), including on strong baselines (Oat-Zero, PRIME-Zero, RePO-Zero).

Stabilizes weak models: The method prevents collapse in Llama-3.1 8B, a previously unstable model under on-policy RLVR.

Open-sourcing & reproducibility: Authors commit to releasing code and weights and provide training details and an extensible GitHub repo.

**Weaknesses:**

Although ExGRPO effectively improves efficiency and stability in RLVR training, it remains heuristic?. The method is also limited to verifiable reasoning tasks and has not been validated on open-ended domains like dialogue or code generation.

**Questions:**

How would ExGRPO perform on tasks with non-verifiable or sparse rewards, such as creative reasoning or open-ended dialogue?

Could the valuable-but-incorrect trajectories (those with correct reasoning but wrong final answers) also be leveraged instead of being discarded?

---

> ### Author Response · Authors · 2025-11-25
> **Reply to Reviewer wSTN (1/2)**
>
> Thank you for the insightful comments. We address the concerns by showing how ExGRPO can be extended to non-verifiable RL settings (W.1 & Q.1), and clarifying its implicit handling of valuable-but-incorrect trajectories (Q.2). Detailed point-by-point responses follow below.
>
> ---
>
> > **Weakness 1. & Question 1.** Although ExGRPO effectively improves efficiency and stability in RLVR training, it remains heuristic?. The method is also limited to verifiable reasoning tasks and has not been validated on open-ended domains like dialogue or code generation.
> How would ExGRPO perform on tasks with non-verifiable or sparse rewards, such as creative reasoning or open-ended dialogue?
> >
>
> **Response:**
>
> We agree that extending ExGRPO to RL with continuous, preference-model rewards is an important direction. Our method can be adapted by redefining the notion of difficulty without relying on binary correctness.
>
> In preference-reward settings, we replace the original correctness metric with the within-query reward variance computed across multiple rollouts. This variance provides a natural proxy for difficulty, while trajectory entropy remains applicable as a measure of reasoning validity.
>
> Specifically, for each query $q$ with $K$ rollouts, we compute the reward standard deviation $\sigma_q$. Since $\sigma_q$ has no intrinsic scale, we map it to a discrete difficulty bucket using relative, batch-wise normalization. Given all variances $\{\sigma_1, \ldots, \sigma_M\}$ within the current batch, we apply a linear min–max normalization, $\tilde{\sigma}_q$ =
>
> $\frac{\sigma_q - \sigma_{\min}}{\max(\sigma_{\max} - \sigma_{\min})}$, and then discretize $\tilde{\sigma}_q \in [0,1]$ into {$0, \ldots, K$} via round$(\tilde{\sigma}_q \cdot K)$.
>
> Each query is then assigned to its corresponding bucket and updated using the ExGRPO mechanism originally developed for the binary-correctness setting.
>
> Under this formulation, buckets still preserve the nature of discrete difficulty levels. As a result, ExGRPO’s sampling mechanism, which was previously centered around medium success rates, now preferentially selects queries with moderate reward variance, i.e., those that are neither overly stable nor excessively noisy. These correspond to intermediate-difficulty queries where learning gains are often most substantial.
>
> To validate feasibility, we conducted a study on 7k WildChat samples [https://huggingface.co/datasets/princeton-nlp/rl_tulu3_wildchat-if_prompts] with on-policy RL training (2 epochs), using Qwen2.5-7B-Base as the testbed. Results (ArenaHard2, IFEval, IFBench) show a consistent improvement of ExGRPO extension:
>
> | Method | ArenaHard2 | IFEval | IFBench | Avg. |
> | --- | --- | --- | --- | --- |
> | GRPO | 15.1 | 26.1 | 18.7 | 20.0 |
> | ExGRPO | **16.4** | **27.9** | **20.7** | **21.7** |
>
> We acknowledge that this extension is preliminary and that alternative designs (e.g., other measures of difficulty) may further improve generalizability. These preliminary results indicate that ExGRPO can indeed be adapted to standard RL with continuous rewards.
>
> We have incorporated the extension experiments and their details into Section 5.3 of the revised paper.

---

> > ### Author Response · Authors · 2025-11-25
> > **Reply to Reviewer wSTN (2/2)**
> >
> > > **Question 2.** Could the valuable-but-incorrect trajectories (those with correct reasoning but wrong final answers) also be leveraged instead of being discarded?
> > >
> >
> > **Response:**
> >
> > We agree that “valuable-but-incorrect” trajectories hold significant potential and may provide useful learning signals, as we discussed this possibility in our Limitations section. In practice, fully leveraging such trajectories remains technically challenging, and our method already captures *part of this signal* in an implicit way.
> >
> > (a) Explicitly leveraging incorrect trajectories is non-trivial. Unlike verifiable final answers, the quality of an incorrect reasoning path varies widely: from near-miss to even misleading. Identifying which failures are “valuable’’ requires a reliable judge of reasoning quality, but existing LLM self-evaluation remains unreliable and is itself an active research challenge [2]. Directly replaying all failed trajectories risks contaminating the training set with low-quality or inconsistent reasoning. This would be an open question for future research.
> >
> > (b) Our method already retains some failure-based learning signals. Although we do not explicitly replay failed trajectories, replayed *successful* trajectories can still yield failures as the policy evolves. A trajectory that was previously solvable may no longer be solved by the current model, and such *re-emergent failures* naturally provide corrective gradients. This mechanism preserves many benefits of “instructive failures’’ without the need to mine or curate failure data [1].
> >
> > Overall, while valuable failed chains can be helpful, reliably isolating them remains open. We view this as a promising future direction, but our current design somewhat captures part of their signal while avoiding the pitfalls of explicitly including noisy failures.
> >
> > **References**
> >
> > [1] Zhu, X., Xia, M., Wei, Z., Chen, W. L., Chen, D., & Meng, Y. (2025). The surprising effectiveness of negative reinforcement in LLM reasoning. *arXiv preprint arXiv:2506.01347*.
> >
> > [2] Feng, Y., Kempe, J., Zhang, C., Jain, P., & Hartshorn, A. (2025). What characterizes effective reasoning? revisiting length, review, and structure of cot. *arXiv preprint arXiv:2509.19284*.

---

### Official Review · Reviewer_B8SA · 2025-10-19

**Soundness:** 3
**Presentation:** 3
**Contribution:** 3
**Rating:** 4
**Confidence:** 3

**Summary:**

This paper addresses the computational inefficiency and instability of standard on-policy Reinforcement Learning from Verifiable Rewards (RLVR) for training large reasoning model. The authors first investigate what constitutes a valuable reasoning experience, identifying rollout correctness (favoring medium-difficulty problems) and trajectory entropy (favoring low-entropy chains) as effective indicators of quality. Based on these insights, they propose ExGRPO (Experiential Group Relative Policy Optimization), a framework that organizes and prioritizes these high-value experiences in a replay buffer. ExGRPO employs a mixed-policy objective that strategically balances fresh exploration with the exploitation of these selected past successes.

**Strengths:**

1. Simple but Clear motivation

2. Simple implementation

3. Clear improvement compared to GRPO

4. Various experiments

**Weaknesses:**

1. Depending on highly heuristic results

**Questions:**

1. It seems difficult to regard a task with 25% accuracy as having the same learning value as one with 75% accuracy. Have you considered adopting a more fine-grained categorization or another prioritization scheme?

2. For each sampled query from the buffer, it is mentioned that the entropy of all stored candidate trajectories needs to be recomputed at every step with respect to the current policy ($\pi_{\theta}$). I’m wondering whether the computational overhead of this process is negligible.

3. To validate the core heuristic of “low entropy,” we used a much more powerful model, Qwen3-32B, as an external judge compared to the 7B target model. I’m curious whether similar patterns would emerge with other models — does this imply that a significantly more capable “teacher” model is required?

---

> ### Author Response · Authors · 2025-11-25
> **Reply to Reviewer B8SA (1/2)**
>
> Thank you for your thoughtful review, particularly regarding the validation of our heuristics and computational efficiency. In response, we have added comprehensive ablation studies to justify our design choices (W.1, Q.1) and provided a detailed runtime analysis (Q.2) to demonstrate the practicality of our method. Detailed point-by-point responses follow below.
>
> ---
>
> > **Weakness 1.** Depending on highly heuristic results.
>
>
> **Response:**
>
> We appreciate this comment and would like to clarify the rationale behind our design.
> While the criteria for experience selection (i.e., prioritizing medium-difficulty questions and low-entropy trajectories) are indeed empirically motivated, they are derived directly from the comprehensive analysis in Section 3.2 rather than arbitrary choices. Furthermore, the underlying framework itself is principled, with a theoretical analysis of the objective provided in the Appendix. We believe the combination of analysis-driven selection criteria (identifying what to learn from) and a principled optimization framework (defining how to learn) constitutes a  strength of our approach compared to previous study.
> To further validate these design choices, we conducted additional ablation studies comparing our approach against alternative bucket sampling distributions and trajectory selection strategies.
>
> **(a) Comparison of Bucketing Sampling Strategies**
>
> Our method employs a probabilistic sampling method based on a Gaussian distribution $p \propto \mathcal{N}(\text{Acc}(q^*); \mu=0.5, \sigma)$, which prioritizes buckets around 50% correctness without using hard thresholds. We performed a sensitivity analysis by varying the mean ($\mu$) and variance ($\sigma$) of the Gaussian distribution used for bucket sampling. We compared our default setting ($\mu=0.5, \sigma=1$) with: (i) a wider focus ($\sigma=1.5$), (ii) a narrower focus ($\sigma=0.5$), and (iii) a harder-biased distribution ($\mu=0.25$, as also suggested in Question 1).
>
> | **Sampling Strategy** | **Settings** | **AIME24** | **AIME25** | **AMC** | **MATH** | **Minerva** | **Olym.** | **A. (ID)** | **ARC-c** | **GPQA*** | **MMLU-P** | **A. (OOD)** |
> | --- | --- | --- | --- | --- | --- | --- | --- | --- | --- | --- | --- | --- |
> | On-Policy | — | 24.9 | 15.5 | 59.2 | 84.8 | 38.2 | 49.3 | 45.3 | 82.6 | 37.4 | 49.2 | 56.4 |
> | **ExGRPO (Vanilla)** | **μ=0.5, σ=1** | **31.6** | 18.7 | **66.3** | **87.4** | 36.0 | **50.1** | **48.4** | **84.7** | 37.4 | **52.9** | **58.3** |
> | Wider Focus | σ=1.5 | 22.0 | **19.2** | 62.3 | 85.4 | **38.2** | 49.9 | 46.2 | 81.7 | **44.4** | 48.6 | 58.2 |
> | Narrower Focus | σ=0.5 | 24.9 | 17.4 | 64.9 | 86.6 | 34.2 | 49.0 | 46.2 | 72.9 | 40.4 | 51.1 | 54.8 |
> | Hard-Biased | μ=0.25 | 22.4 | 19.6 | 61.9 | 86.0 | 38.2 | 49.8 | 46.3 | 83.6 | 41.4 | 49.4 | 58.2 |
>
> *(“A.” denotes average; “MATH” is MATH-500; “Olym.” is OlympiadBench.)*
>
> **(b) Comparison of Trajectory Selection Strategies**
> We also compared our low-entropy selection against other strategies, including highest-entropy selection and random selection.
>
> | Selection Strategy | **AIME24** | **AIME25** | **AMC** | **MATH** | **Minerva** | **Olym.** | **A. (ID)** | **ARC-c** | **GPQA*** | **MMLU-P** | **A. (OOD)** |
> | --- | --- | --- | --- | --- | --- | --- | --- | --- | --- | --- | --- |
> | Lowest-Entropy  | **31.6** | **18.7** | **66.3** | **87.4** | 36.0 | 50.1 | **48.4** | **84.7** | 37.4 | **52.9** | 58.3 |
> | Highest-Entropy | 25.0 | 16.0 | 62.9 | 84.6 | **36.4** | **51.0** | 46.0 | 81.9 | **49.5** | 50.1 | **60.5** |
> | Random | 26.0 | 17.0 | 63.7 | 84.8 | 35.7 | 50.7 | 46.3 | 82.3 | 41.9 | 51.6 | 58.6 |
>
> Overall, the results demonstrate that our proposed heuristics (medium-difficulty focus and lowest-entropy trajectory selection) consistently outperform alternative configurations on average ID performance, confirming the validity of our analysis-driven design. And for all the tasks, 6 out of 9 wins, reflecting the generalization scope.
>
> We have added this ablation study to Section 6.2 and Appendix E in the revised paper to justify our design. We appreciate your suggestion, as it has strengthened our empirical analysis.

---

> > ### Author Response · Authors · 2025-11-25
> > **Reply to Reviewer B8SA (2/2)**
> >
> > > **Question 1.** It seems difficult to regard a task with 25% accuracy as having the same learning value as one with 75% accuracy. Have you considered adopting a more fine-grained categorization or another prioritization scheme?
> > >
> >
> > **Response:**
> >
> > This is an insightful point regarding the symmetry of learning value. To empirically validate this, we tested a Hard-Biased setting ($\mu=0.25$), which shifts the sampling focus toward harder tasks (25% accuracy) while reducing the probability of revisiting easier ones (75% accuracy). As shown in the table above (in the response to Weakness 1), this strategy yields inferior results compared to the symmetric medium-difficulty focus ($\mu=0.5$). The drop in Average ID performance suggests that treating 25% and 75% accuracy symmetrically provides a better balance between learning harder tasks and consolidating knowledge of easier tasks.
> >
> > ---
> >
> > > **Question 2.** For each sampled query from the buffer, it is mentioned that the entropy of all stored candidate trajectories needs to be recomputed at every step with respect to the current policy. I’m wondering whether the computational overhead of this process is negligible.
> > >
> >
> > **Response:**
> >
> > The computational overhead is manageable because entropy is not recomputed for the entire buffer. The process operates on a per-batch basis: First, we sample a batch of queries from a specific difficulty bucket. Then, for only these sampled queries, we retrieve all their associated successful trajectories from the buffer. Finally, we calculate the entropy for this small subset of trajectories (typically a few dozen per query) with respect to the current policy and select the lowest-entropy one.
> >
> > We benchmarked the actual training time and memory usage against the baseline (without experience replay) across two model sizes and GPU environments. As shown, the average time overhead is approximately 19% for the 7B model and negligible (~2%) for the 1.5B model, with only a marginal increase in memory footprint. For modern training hardware, the increased costs are acceptable.
> >
> > | Model & Hardware | Method | Replay Buffer RAM | Min. Time / Step (s) | Avg. Time / Step (s) |
> > | --- | --- | --- | --- | --- |
> > | **Qwen2.5-Math-7B (8 GPUs)** | DR.GRPO (On-policy Baseline) | – | 139.83 | 155.38 |
> > |  | ExGRPO (Ours) | 1.77 GB | 151.36 | 184.92 |
> > |  | **Overhead** | **+1.77 GB** | **+8.2%** | **+19.0%** |
> > | **Qwen2.5-Math-1.5B (4 GPUs)** | DR.GRPO (On-policy Baseline) | – | 150.55 | 185.00 |
> > |  | ExGRPO (Ours) | 1.05 GB | 163.92 | 188.85 |
> > |  | **Overhead** | **+1.05 GB** | **+8.9%** | **+2.1%** |
> >
> > We have included a detailed analysis in the updated paper version, with all newly added contents (Appendix E.5) highlighted in blue.
> >
> > ---
> >
> > > **Question 3.** To validate the core heuristic of “low entropy,” we used a much more powerful model, Qwen3-32B, as an external judge compared to the 7B target model. I’m curious whether similar patterns would emerge with other models — does this imply that a significantly more capable “teacher” model is required?
> > >
> >
> > **Response:**
> >
> > We would like to clarify that the stronger judge model was used only in the initial analytical study (Section 3.2). The purpose was to establish a "ground truth" for what constitutes a correct reasoning path, independent of the capabilities of the smaller model being trained. This allowed us to validate the heuristic that low-entropy paths are generally more valuable.
> >
> > However, during the actual training of ExGRPO, no external teacher model is used. The entropy of each trajectory is always calculated with respect to the current policy of the model being trained. The method is entirely self-contained.
> >
> > We have revised Section 4.1 to make this distinction absolutely clear and avoid any confusion.

---

> > > ### Comment · Reviewer_B8SA · 2025-11-26
> > >
> > > Thank you very much for the sufficient and clear explanation.
> > >
> > > The clear justification has successfully resolved my concerns (especially those related to Question 1).
> > >
> > > Upon considering the outperforming performance of this method, I recognize that I previously undervalued this contribution.
> > >
> > > Accordingly, I will update my evaluation to reflect the substantial clarity and significance demonstrated by the authors.

---

> > > > ### Author Response · Authors · 2025-11-27
> > > >
> > > > Thank you very much for your constructive and timely feedback. We are glad to hear that our response resolved your concerns. Your review has helped us improve the quality of our paper, and we are grateful for your recognition of our contribution and the updated evaluation.

---

### Official Review · Reviewer_o61x · 2025-10-30

**Soundness:** 3
**Presentation:** 3
**Contribution:** 3
**Rating:** 6
**Confidence:** 3

**Summary:**

The paper introduces ExGRPO (Experiential Group Relative Policy Optimization), a reinforcement learning framework designed to enhance the reasoning ability of large language models (LLMs) under the Reinforcement Learning with Verifiable Rewards (RLVR) paradigm. Unlike conventional on-policy RLVR methods that discard rollout experiences after each update, ExGRPO reuses past successful experiences through a structured experience management and replay mechanism. The framework partitions experiences by rollout correctness and trajectory entropy, selecting medium-difficulty, low-entropy samples for replay. Experiments across multiple backbones (1.5B–8B models, including Qwen and Llama families) demonstrate consistent improvements in mathematical reasoning performance.

**Strengths:**

1. Quality: The empirical dynamics of the algorithm are well investigated by the analysis in Section 6, where theoretical justifications are also provided to ensure the unbiasedness of the proposed formula in Equation 4. Experimental settings follow the standard setup in this field.

2. Clarity: This paper is well written and easy to follow.

3. Novelty: The proposed experience replay management and its combination with on-policy training look new to me, though the accuracy- and entropy-based approach is not new, given the existence of similar literature [1][2] in the past.

**Weaknesses:**

1. Significance:
  - The selection criteria depend on the metric of correctness, which requires the reward signal to be verifiable. This introduces additional challenges when extending to more general open-domain tasks.
  - While ExGRPO achieves better performance than the on-policy baseline, it also incurs additional time and memory costs for accuracy recomputation and experience storage. It is worth discussing whether those costs will pose practical challenges or if there are trade-offs between cost and performance.

2. Quality:
  - More baselines are encouraged to be compared with, such as [1][2][3].

  - Minor issues:
    + line 276: typo: "a advantage group" -> "an advantage group"

### References
[1]: Zhang, Jixiao, and Chunsheng Zuo. "Grpo-lead: A difficulty-aware reinforcement learning approach for concise mathematical reasoning in language models." arXiv preprint arXiv:2504.09696 (2025).

[2]: Chen, Minghan, et al. "Seed-grpo: Semantic entropy enhanced grpo for uncertainty-aware policy optimization." arXiv preprint arXiv:2505.12346 (2025).

[3]: Wei Xiong, et al. A minimalist approach to llm reasoning: from rejection
sampling to reinforce. arXiv preprint arXiv:2504.11343, 2025.

[4]: Qiying Yu, et al. Dapo: An open-source llm reinforcement learning system at scale. arXiv preprint arXiv:2503.14476, 2025

**Questions:**

* line 258: The choice of the Gaussian distribution seems heuristic here. I am wondering if there is any theoretical justification or empirical evidence to support this choice.

---

> ### Author Response · Authors · 2025-11-25
> **Reply to Reviewer o61x (1/2)**
>
> Thank you for your careful review. In this response, we demonstrate ExGRPO’s generalizability to continuous preference rewards (W1.1), quantify its minimal computational overhead (W1.2), and empirically validate the robustness of our Gaussian sampling design (Q1).
>
> Detailed point-by-point responses follow below.
>
> ---
>
> > **Weakness 1.1.** The selection criteria depend on the metric of correctness, which requires the reward signal to be verifiable. This introduces additional challenges when extending to more general open-domain tasks.
> >
>
> **Response:**
>
> We agree that extending ExGRPO to RL with continuous, preference-model rewards is an important direction. Our method can be adapted by redefining the notion of difficulty without relying on binary correctness.
>
> In preference-reward settings, we replace the original correctness metric with the within-query reward variance computed across multiple rollouts. This variance provides a natural proxy for difficulty, while trajectory entropy remains applicable as a measure of reasoning validity.
>
> Specifically, for each query $q$ with $K$ rollouts, we compute the reward standard deviation $\sigma_q$. Since $\sigma_q$ has no intrinsic scale, we map it to a discrete difficulty bucket using relative, batch-wise normalization. Given all variances $\{\sigma_1, \ldots, \sigma_M\}$ within the current batch, we apply a linear min–max normalization, $\tilde{\sigma}_q$ =
>
> $\frac{\sigma_q - \sigma_{\min}}{\max(\sigma_{\max} - \sigma_{\min})}$, and then discretize $\tilde{\sigma}_q \in [0,1]$ into {$0, \ldots, K$} via round$(\tilde{\sigma}_q \cdot K)$.
>
> Under this formulation, buckets still preserve the nature of discrete difficulty levels. As a result, ExGRPO’s sampling mechanism, which was previously centered around medium success rates, now preferentially selects queries with moderate reward variance. These correspond to intermediate-difficulty queries where learning gains are often most substantial.
>
> To validate feasibility, we conducted a study on a 7k WildChat-IF subset [https://huggingface.co/datasets/princeton-nlp/rl_tulu3_wildchat-if_prompts] with on-policy RL training (2 epochs), using Qwen2.5-7B-Base as the base model and Skywork-Reward-Llama-3.1-8B-v0.2 as the reward model. Results (ArenaHard2, IFEval, IFBench) show a consistent improvement of ExGRPO extension:
>
> | Model | ArenaHard2 | IFEval | IFBench | Avg. |
> | --- | --- | --- | --- | --- |
> | GRPO | 15.1 | 26.1 | 18.7 | 20.0 |
> | ExGRPO | **16.4** | **27.9** | **20.7** | **21.7** |
>
> We acknowledge that this extension is preliminary and that alternative designs (e.g., other measures of difficulty) may further improve generalizability. These preliminary results indicate that ExGRPO can indeed be adapted to standard RL with continuous rewards.
>
> We have incorporated the extension experiments and their details into Section 5.3 of the revised paper.
>
> ---
>
> > **Weakness 1.2.** While ExGRPO achieves better performance than the on-policy baseline, it also incurs additional time and memory costs for accuracy recomputation and experience storage. It is worth discussing whether those costs will pose practical challenges or if there are trade-offs between cost and performance.
> >
>
> **Response:**
>
> Regarding **memory usage**, the replay buffer does add to memory requirements. However, this is a manageable trade-off for experience replay methods. In our experiments, a buffer storing 202,011 trajectories and log-probability status (this is maximum usage in our experiment for Qwen2.5-Math-7B as measured at the end of training) required approximately 1.77 GB of RAM, which is a reasonable footprint on modern training hardware.
>
> Regarding **computational throughput**, the main computational overhead comes from re-calculating trajectory entropies. To clarify the process also raised in Question 1, we **do not re-calculate entropy for all trajectories** in the buffer at every step. Instead, the process is efficient and parallelizable. We sample a batch of queries from a specific difficulty bucket, retrieve their successful trajectories from the buffer, and only calculate the entropy for this small subset of trajectories. This computation incurs negligible training overhead.
>
> To empirically test it, we recorded the statistics spanning on different model scales and devices, are shown below:
>
> | Model & Hardware | Method | Replay Buffer RAM | Min. Time / Step (s) | Avg. Time / Step (s) |
> | --- | --- | --- | --- | --- |
> | **Qwen2.5-Math-7B (8 GPUs)** | DR.GRPO (On-policy Baseline) | – | 139.83 | 155.38 |
> |  | ExGRPO (Ours) | 1.77 GB | 151.36 | 184.92 |
> |  | **Overhead** | **+1.77 GB** | **+8.2%** | **+19.0%** |
> | **Qwen2.5-Math-1.5B (4 GPUs)** | DR.GRPO (On-policy Baseline) | – | 150.55 | 185.00 |
> |  | ExGRPO (Ours) | 1.05 GB | 163.92 | 188.85 |
> |  | **Overhead** | **+1.05 GB** | **+8.9%** | **+2.1%** |
>
> We have included a detailed analysis in the updated paper version, with all newly added contents (Appendix E.5) highlighted in blue.

---

> > ### Author Response · Authors · 2025-11-25
> > **Reply to Reviewer o61x (2/2)**
> >
> > > **Weakness 2.1.** More baselines are encouraged to be compared with, such as [1][2][3].
> >
> >
> > **Response:**
> >
> > We appreciate the pointers to these relevant works. While they share some general concepts, ExGRPO differs in its detailed mechanism and focus:
> >
> > - GRPO-Lead [1] adjusts advantage reweighting based on online success rates. In contrast, ExGRPO only uses online success rates to group the past experience items into different sampling buckets.
> > - SEED-GRPO [2] uses semantic entropy to modulate the magnitude of policy updates. ExGRPO, however, uses entropy for trajectory selection for a specific question, making our contribution orthogonal to gradient modulation techniques.
> > - Reinforce-Rej [3] focuses on filtering entirely online correct/incorrect prompts. ExGRPO’s retired set filter out masted historical items to facilitate experience-based learning.
> >
> > We have added a detailed discussion of these connections in the revised paper (Section 3.2 & 4.1). Given the short rebuttal period, we prioritized the extensive experiments for W1.1 and Q1. However, we believe ExGRPO offers a distinct contribution by formally introducing principled experience replay to GRPO.
> >
> > **References**
> >
> > [1] Zhang, Jixiao, and Chunsheng Zuo. (2025). Grpo-lead: A difficulty-aware reinforcement learning approach for concise mathematical reasoning in language models. arXiv preprint arXiv:2504.09696.
> >
> > [2] Chen, Minghan, et al. (2025). Seed-grpo: Semantic entropy enhanced grpo for uncertainty-aware policy optimization." arXiv preprint arXiv:2505.12346.
> >
> > [3] Wei Xiong, et al. (2025). A minimalist approach to llm reasoning: from rejection sampling to reinforce. arXiv preprint arXiv:2504.11343.
> >
> > ---
> >
> > > **Weakness 2.2.** Minor issues:
> > >
> > > - line 276: typo: "a advantage group" -> "an advantage group"
> >
> > **Response:**
> >
> > We have corrected the typo "a advantage group" to "an advantage group." in the revised paper. Thank you for your careful review.
> >
> > ---
> >
> > > **Question 1.** line 258: The choice of the Gaussian distribution seems heuristic here. I am wondering if there is any theoretical justification or empirical evidence to support this choice.
> >
> >
> > **Response:**
> >
> > The Gaussian-based sampling scheme is indeed a heuristic, but it aligns with our analytical finding that queries around 50% correctness provide the most valuable learning signal. A Gaussian centered at this region offers a simple and smooth way to prioritize such medium-difficulty problems while still allowing coverage over the full difficulty spectrum.
> >
> > To further validate this choice, we performed an empirical sensitivity study by varying both the mean $\mu$ and variance $\sigma$ of the Gaussian used for bucket sampling. Specifically, we compared our default setting ($\mu=0.5,\sigma=1$) with (i) a wider focus ($\sigma=1.5$), (ii) a narrower focus ($\sigma=0.5$), and (iii) a harder-biased distribution ($\mu=0.25$). The results are summarized below:
> >
> > | **Sampling Strategy** | **Settings** | **AIME24** | **AIME25** | **AMC** | **MATH** | **Minerva** | **Olym.** | **A. (ID)** | **ARC-c** | **GPQA*** | **MMLU-P** | **A. (OOD)** |
> > | --- | --- | --- | --- | --- | --- | --- | --- | --- | --- | --- | --- | --- |
> > | On-Policy | — | 24.9 | 15.5 | 59.2 | 84.8 | 38.2 | 49.3 | 45.3 | 82.6 | 37.4 | 49.2 | 56.4 |
> > | **ExGRPO (Vanilla)** | **μ=0.5, σ=1** | **31.6** | 18.7 | **66.3** | **87.4** | 36.0 | **50.1** | **48.4** | **84.7** | 37.4 | **52.9** | **58.3** |
> > | Wider Focus | σ=1.5 | 22.0 | **19.2** | 62.3 | 85.4 | **38.2** | 49.9 | 46.2 | 81.7 | **44.4** | 48.6 | 58.2 |
> > | Narrower Focus | σ=0.5 | 24.9 | 17.4 | 64.9 | 86.6 | 34.2 | 49.0 | 46.2 | 72.9 | 40.4 | 51.1 | 54.8 |
> > | Hard-Biased | μ=0.25 | 22.4 | 19.6 | 61.9 | 86.0 | 38.2 | 49.8 | 46.3 | 83.6 | 41.4 | 49.4 | 58.2 |
> >
> > *(“A.” denotes average; “MATH” is MATH-500; “Olym.” is OlympiadBench.)*
> >
> > These results provide empirical support that the chosen Gaussian is not arbitrary; rather, emphasizing medium-difficulty queries centered around 50% correctness is a robust and effective design choice.
> >
> > We have added this ablation study to Section 6.2 in the revised paper to justify our design. We appreciate your suggestion, as it has strengthened our empirical analysis.

---

### Official Review · Reviewer_7NcB · 2025-10-31

**Soundness:** 3
**Presentation:** 3
**Contribution:** 3
**Rating:** 6
**Confidence:** 3

**Summary:**

This paper addresses the inefficiency of on-policy RLVR for LLM reasoning, where valuable experience is wasted. The authors first identify that intermediate-difficulty problems and low-entropy or high-certainty trajectories are the most valuable training data. Based on this, they propose ExGRPO, a framework with principled experience replay. It maintains a buffer, groups experiences into buckets by real-time correctness, and prioritizes sampling medium-difficulty tasks, selecting the lowest-entropy trajectory for replay. Experiments show ExGRPO significantly outperforms on-policy baselines, especially in out-of-distribution generalization and training stability.

**Strengths:**

1. The paper's primary strength is its methodical approach. It doesn't just propose experience replay; it first runs a diagnostic study in Section 3.2 to prove that medium-difficulty questions and low-entropy trajectories are the most valuable. This provides a strong, data-driven empirical foundation for the ExGRPO framework's design.

2. The average gain of +7.6 OOD points across all models is significant. It suggests that learning from a curated buffer of past successes helps the model form more robust and generalizable reasoning patterns compared to noisy on-policy exploration, which may overfit to in-distribution data.

3. The paper's novelty lies in the principled application of experience replay to RLVR. The idea of dynamically bucketing experiences by online correctness and then selecting by trajectory entropy is a new and intelligent combination of ideas that goes beyond naive replay.

4. The method is not based on an empirical heuristic. The appendix provides a solid theoretical analysis of the mixed-policy objective, its variance bounds, and the role of policy shaping. This adds significant depth and soundness to the proposed framework.

5. The paper is well-written. It logically builds its argument from the preliminary study to the final method, and its explanations of complex concepts are intuitive. The figures and tables are clean, informative, and effectively support the main claims.

**Weaknesses:**

1. The method introduces a replay buffer and additional computation for entropy calculation and sampling per step. A discussion on the impact on memory usage and overall training throughput is missing, which makes it hard to assess the method's true efficiency trade-offs.

2. The entire method hinges on the "medium-difficulty" bucket, defined as 25%-75% correctness. This range is presented without justification and is not tested in the ablations. It's unclear how sensitive the model's performance is to this definition. A sensitivity analysis, for example, showing results for 30-70 or 40-60, is a critical missing piece.

3. The method exclusively replays low-entropy successes. This strategy risks two problems: a) it discards "instructive failures" which are known to be valuable for learning, and b) it may reduce policy exploration by only reinforcing high-confidence, correct paths, potentially leading to a less diverse policy. The authors do not sufficiently justify this "success-only" design choice over other possibilities.

4. The reliance on a binary "correctness" signal for bucketing makes the method's application to standard RLHF unclear. These tasks use continuous rewards from a preference model, such as for helpfulness, and it's not obvious how to define "difficulty" buckets in that context. This limits the method's generalizability

**Questions:**

1. As the replay buffer grows to millions of items, is the "lowest-entropy" selection (Section 4.1) recalculated for all trajectories in a sampled bucket at every step? If so, this step will become a significant computational bottleneck of this method.
2. How sensitive is the method to the definition of the difficulty buckets? For instance, what happens to performance if the "medium" bucket is defined more narrowly or more broadly?
3. The paper focuses on replaying low-entropy successes. Do the authors experiment with replaying other types of experiences, such as high-entropy successes or even "instructive failures"? Is there a risk of the policy becoming too narrow by only focusing on confident, correct solutions?
4. Regarding the "Retired Set" (Section 4.1): questions are retired after 100% success. Does this risk "catastrophic forgetting" of these skills? Would it be beneficial to periodically reintroduce a small fraction of "Easy" or "Retired" problems back into the training mix to ensure the model retains these capabilities?

---

> ### Author Response · Authors · 2025-11-25
> **Reply to Reviewer 7NcB (1/4)**
>
> Thank you for your constructive feedback. In response, we have conducted extensive additional experiments to address your concerns, including profiling computational efficiency (W.1&Q.1), validating our design choices (W.2, W.3 & Q.2, Q.3, Q.4) via ablations, and demonstrating generalization (W.4) on continuous reward tasks (WildChat). Detailed point-by-point responses follow below.
>
> ---
>
> > **Weakness 1. & Question 1.** The method introduces a replay buffer and additional computation for entropy calculation and sampling per step. A discussion on the impact on memory usage and overall training throughput is missing, which makes it hard to assess the method's true efficiency trade-offs.
> As the replay buffer grows to millions of items, is the "lowest-entropy" selection (Section 4.1) recalculated for all trajectories in a sampled bucket at every step? If so, this step will become a significant computational bottleneck of this method.
>
>
> **Response:**
>
> Regarding *memory usage*, the replay buffer does add to memory requirements. However, this is a manageable trade-off for experience replay methods. In our experiments, a buffer storing 202,011 trajectories and log-probability status (this is maximum usage in our experiment for Qwen2.5-Math-7B as measured at the end of training) required approximately 1.77 GB of RAM, which is a reasonable footprint on modern training hardware.
>
> Regarding *computational throughput*, the main computational overhead comes from re-calculating trajectory entropies. To clarify the process also raised in Question 1, we **do not re-calculate entropy for all trajectories** in the buffer at every step. Instead, the process is much more efficient:
>
> - First, we sample a batch of queries from a specific difficulty bucket.
> - Then, for only these sampled queries, we retrieve all their associated successful trajectories from the buffer.
> - Finally, we calculate the entropy for this small subset of trajectories (typically a few dozen per query) with respect to the current policy and select the lowest-entropy one.
>
> This selective, per-batch computation is parallelizable and has a modest impact on the training time. To empirically test it, we recorded the statistics spanning on different model scales and devices, are shown below:
>
> | Model & Hardware | Method | Replay Buffer RAM | Min. Time / Step (s) | Avg. Time / Step (s) |
> | --- | --- | --- | --- | --- |
> | **Qwen2.5-Math-7B (8 GPUs)** | DR.GRPO (On-policy Baseline) | – | 139.83 | 155.38 |
> |  | ExGRPO (Ours) | 1.77 GB | 151.36 | 184.92 |
> |  | **Overhead** | **+1.77 GB** | **+8.2%** | **+19.0%** |
> | **Qwen2.5-Math-1.5B (4 GPUs)** | DR.GRPO (On-policy Baseline) | – | 150.55 | 185.00 |
> |  | ExGRPO (Ours) | 1.05 GB | 163.92 | 188.85 |
> |  | **Overhead** | **+1.05 GB** | **+8.9%** | **+2.1%** |
>
> We have included a detailed analysis in the updated paper version, with all newly added contents (Appendix E.5) highlighted in blue.

---

> > ### Author Response · Authors · 2025-11-25
> > **Reply to Reviewer 7NcB (2/4)**
> >
> > > **Weakness 2. & Question 2.** The entire method hinges on the "medium-difficulty" bucket, defined as 25%-75% correctness. This range is presented without justification and is not tested in the ablations. It's unclear how sensitive the model's performance is to this definition. A sensitivity analysis, for example, showing results for 30-70 or 40-60, is a critical missing piece.
> > How sensitive is the method to the definition of the difficulty buckets? For instance, what happens to performance if the "medium" bucket is defined more narrowly or more broadly?
> > >
> >
> > **Response:**
> >
> > We wish to clarify that the deterministic 25%-75% range mentioned was used only for the analytical study. For the actual training (as detailed in Section 4.1), we imply a probabilistic sampling method based on a Gaussian distribution $p \propto \mathcal{N}(\text{Acc}(q^*); \mu=0.5, \sigma)$, which prioritizes buckets around 50% correctness without using hard thresholds. This soft-prioritization naturally assigns higher probabilities to medium-difficulty examples without defining a rigid range.
> >
> > We agree, however, that a sensitivity analysis is crucial to demonstrate robustness. In our method, the standard deviation $\sigma$ of the Gaussian sampler directly controls the "width" of prioritization. This is analogous to your suggestion of testing different ranges (e.g., a smaller $\sigma$ for a 40-60-like focus, and a larger $\sigma$ for a broader focus).
> >
> > As suggested, we performed a sensitivity analysis by varying $\sigma$ (comparing our default $\sigma=1$ against a narrower $\sigma=0.5$ and a wider $\sigma=1.5$). The results are presented below:
> >
> > | **Sampling Strategy** | **$\sigma$** | **AIME24** | **AIME25** | **AMC** | **MATH** | **Minerva** | **Olym.** | **A. (ID)** | **ARC-c** | **GPQA*** | **MMLU-P** | **A. (OOD)** |
> > | --- | --- | --- | --- | --- | --- | --- | --- | --- | --- | --- | --- | --- |
> > | On-Policy | N/A | 24.9 | 15.5 | 59.2 | 84.8 | 38.2 | 49.3 | 45.3 | 82.6 | 37.4 | 49.2 | 56.4 |
> > | ExGRPO (Vanilla) | **1.0** | **31.6** | 18.7 | **66.3** | **87.4** | 36.0 | **50.1** | **48.4** | **84.7** | 37.4 | **52.9** | **58.3** |
> > | Wider Focus | 1.5 | 22.0 | **19.2** | 62.3 | 85.4 | **38.2** | 49.9 | 46.2 | 81.7 | **44.4** | 48.6 | 58.2 |
> > | Narrower Focus | 0.5 | 24.9 | 17.4 | 64.9 | 86.6 | 34.2 | 49.0 | 46.2 | 72.9 | 40.4 | 51.1 | 54.8 |
> >
> > *(Note: "MATH" refers to MATH-500; "Olym." to OlympiadBench; “A.” to Average score.)*
> >
> > The results indicate that the method is relatively robust, but performance peaks around $\sigma=1$. Specifically:
> >
> > **For Wider Focus ($\sigma=1.5$)**, a broader distribution maintains better general capabilities (e.g., higher GPQA scores) but dilutes the efficiency on core math reasoning tasks (AIME24 drops from 31.6 to 22.0), confirming that prioritizing medium-difficulty samples is indeed more data-efficient than a uniform-like distribution.
> >
> > **For Narrower Focus ($\sigma=0.5$)**, while performance remains competitive on some math benchmarks (e.g., AMC), the model suffers degradation on general reasoning tasks like ARC-c (dropping from 84.7 to 72.9). This suggests that overly focusing on the "learning frontier" may lead to loss of diversity in training data.
> >
> > We have added this ablation study to Section 6.2 in the revised paper to justify our design. We appreciate your suggestion, as it has strengthened our empirical analysis.

---

> ### Author Response · Authors · 2025-11-25
> **Reply to Reviewer 7NcB (3/4)**
>
> > **Weakness 3. & Question 3.** The method exclusively replays low-entropy successes. This strategy risks two problems: a) it discards "instructive failures" which are known to be valuable for learning, and b) it may reduce policy exploration by only reinforcing high-confidence, correct paths, potentially leading to a less diverse policy. The authors do not sufficiently justify this "success-only" design choice over other possibilities.
> The paper focuses on replaying low-entropy successes. Do the authors experiment with replaying other types of experiences, such as high-entropy successes or even "instructive failures"? Is there a risk of the policy becoming too narrow by only focusing on confident, correct solutions?
>
>
> **Response:**
>
> (a) On the role of “instructive failures.” We agree that “instructive failures” can be valuable for learning, as also noted in our Limitations section. However, our method does not remove all failure-based learning signals because *replayed successes can still produce failures*. Our method does not entirely discard the learning signal from failures. We replay trajectories that were *previously* logged as successes, but the evolving policy may no longer solve these problems consistently. Such *re-emergent failures* on known-solvable tasks provide natural corrective signals. This mechanism brings many of the benefits of “instructive failures” without explicitly mining or labeling them [1].
>
> *On the other hand, identifying truly instructive failures remains a challenge.* Unlike verifiable outcomes, a failed reasoning chain can range from a valuable near-miss to a completely uninformative trajectory. Without a trusted external judge, asking the model to self-judge the "instructiveness" of its own flawed reasoning is notoriously difficult and an open research problem [2]. Explicitly replaying failures risks polluting the training data with low-quality or misleading examples.
>
> Below we clarify why our method focuses on low-entropy successes and why this choice does not harm exploration or generalization:
>
> (b) On exploration and policy diversity. We argue that our method does not completely reduce exploration.  As detailed in Section 4.2,  the policy shaping technique is explicitly designed to maintain exploration, which is reflected the sustained policy entropy throughout training (Figure 8, Appendix). Furthermore, our mixed objective includes fresh on-policy rollouts, ensuring continual exploration and preventing the model from overfitting to a narrow set of high-confidence trajectories.
>
> (c) Additional experiments on alternative replay strategies. To empirically address this question, we ran additional experiments (some of which are detailed in Appendix E.4) replaying other types of experiences, such as high-entropy successes and random selection. As shown in the table below, the Lowest-Entropy strategy consistently performs best on most benchmarks, indicating that replaying low-entropy reasoning traces effectively consolidates reasoning ability while preserving generalization (for all the tasks, 6 out of 9 wins).
>
> | Selection Strategy | **AIME24** | **AIME25** | **AMC** | **MATH** | **Minerva** | **Olym.** | **A. (ID)** | **ARC-c** | **GPQA*** | **MMLU-P** | **A. (OOD)** |
> | --- | --- | --- | --- | --- | --- | --- | --- | --- | --- | --- | --- |
> | Lowest-Entropy  | **31.6** | **18.7** | **66.3** | **87.4** | 36.0 | 50.1 | **48.4** | **84.7** | 37.4 | **52.9** | 58.3 |
> | Highest-Entropy | 25.0 | 16.0 | 62.9 | 84.6 | **36.4** | **51.0** | 46.0 | 81.9 | **49.5** | 50.1 | **60.5** |
> | Random | 26.0 | 17.0 | 63.7 | 84.8 | 35.7 | 50.7 | 46.3 | 82.3 | 41.9 | 51.6 | 58.6 |
>
> *(Note: "MATH" refers to MATH-500; "Olym." to OlympiadBench; “A.” to Average score.)*
>
> **References**
>
> [1] Zhu, X., Xia, M., Wei, Z., Chen, W. L., Chen, D., & Meng, Y. (2025). The surprising effectiveness of negative reinforcement in LLM reasoning. *arXiv preprint arXiv:2506.01347*.
>
> [2] Feng, Y., Kempe, J., Zhang, C., Jain, P., & Hartshorn, A. (2025). What characterizes effective reasoning? revisiting length, review, and structure of cot. *arXiv preprint arXiv:2509.19284*.

---

> ### Author Response · Authors · 2025-11-25
> **Reply to Reviewer 7NcB (4/4)**
>
> > **Weakness 4.** The reliance on a binary "correctness" signal for bucketing makes the method's application to standard RLHF unclear. These tasks use continuous rewards from a preference model, such as for helpfulness, and it's not obvious how to define "difficulty" buckets in that context. This limits the method's generalizability.
> >
>
> **Response:**
>
> We agree that extending ExGRPO to RL with continuous, preference-model rewards is an important direction. Our method can be adapted by redefining the notion of difficulty without relying on binary correctness.
>
> In preference-reward settings, we replace the original correctness metric with the within-query reward variance computed across multiple rollouts. This variance provides a natural proxy for difficulty, while trajectory entropy remains applicable as a measure of reasoning validity.
>
> Each query is then assigned to its corresponding bucket and updated using the ExGRPO mechanism originally developed for the binary-correctness setting.
>
> Specifically, for each query $q$ with $K$ rollouts, we compute the reward standard deviation $\sigma_q$. Since $\sigma_q$ has no intrinsic scale, we map it to a discrete difficulty bucket using relative, batch-wise normalization. Given all variances $\{\sigma_1, \ldots, \sigma_M\}$ within the current batch, we apply a linear min–max normalization, $\tilde{\sigma}_q$ =
>
> $\frac{\sigma_q - \sigma_{\min}}{\max(\sigma_{\max} - \sigma_{\min})}$, and then discretize $\tilde{\sigma}_q \in [0,1]$ into {$0, \ldots, K$} via round$(\tilde{\sigma}_q \cdot K)$.
>
> Under this formulation, buckets still preserve the nature of discrete difficulty levels. As a result, ExGRPO’s sampling mechanism, which was previously centered around medium success rates, now preferentially selects queries with moderate reward variance, i.e., those that are neither overly stable nor excessively noisy. These correspond to intermediate-difficulty queries where learning gains are often most substantial.
>
> To validate feasibility, we conducted a study on a 7k WildChat-IF subset [https://huggingface.co/datasets/princeton-nlp/rl_tulu3_wildchat-if_prompts] with on-policy RL training (2 epochs), using Qwen2.5-7B-Base as the base model and Skywork-Reward-Llama-3.1-8B-v0.2 as the reward model. Results (ArenaHard2, IFEval, IFBench) show a consistent improvement of ExGRPO extension:
>
> | Method | ArenaHard2 | IFEval | IFBench | Avg. |
> | --- | --- | --- | --- | --- |
> | GRPO | 15.1 | 26.1 | 18.7 | 20.0 |
> | ExGRPO | **16.4** | **27.9** | **20.7** | **21.7** |
>
> We acknowledge that this extension is preliminary and that alternative designs (e.g., other measures of difficulty) may furtherash improve generalizability. These preliminary results indicate that ExGRPO can indeed be adapted to standard RL with continuous rewards.
>
> We have incorporated the extension experiments and their details into Section 5.3 of the revised paper.
>
> ---
>
> > **Question 4.** Regarding the "Retired Set" (Section 4.1): questions are retired after 100% success. Does this risk "catastrophic forgetting" of these skills? Would it be beneficial to periodically reintroduce a small fraction of "Easy" or "Retired" problems back into the training mix to ensure the model retains these capabilities?
> >
>
> **Response:**
>
> This is a insightful question. We conducted an additional experiment to evaluate whether periodically reintroducing “Retired” (i.e., 100%-success) queries helps maintain previously acquired skills. In this experiment, every 25 training steps we sampled a small fraction (10% of the batch size) of queries from the Retired set and injected them back into the training mix. These reintroduced queries replaced an equal number of samples drawn from the experience replay buffer, while the rest of the training pipeline remained unchanged.
>
> Based on the results shown in Table, reintroducing retired queries degraded performance. We hypothesize that since these problems are already fully mastered, re-training on them wastes valuable compute budget that could otherwise be spent on "frontier" (medium-difficulty) problems. The model appears to retain these skills sufficiently without explicit review (likely due to shared underlying reasoning patterns).
>
> | Replay Strategy | **AIME24** | **AIME25** | **AMC** | **MATH** | **Minerva** | **Olym.** | **A. (ID)** | **ARC-c** | **GPQA*** | **MMLU-P** | **A. (OOD)** |
> | --- | --- | --- | --- | --- | --- | --- | --- | --- | --- | --- | --- |
> | ExGRPO | 31.6 | 18.7 | 66.3 | 87.4 | 36.0 | 50.1 | 48.4 | 84.7 | 37.4 | 52.9 | 58.3 |
> | w/ Retried Review | 23.2 | 17.7 | 61.0 | 84.0 | 34.9 | 50.7 | 45.3 | 81.7 | 41.4 | 48.0 | 57.1 |
>
> We have added this ablation study to Appendix E.4 in the revised paper.

---

### Comment · Reviewer_wSTN · 2025-11-27
**Respond to author's response**

My questions are well sovled.

---

### Author Response · Authors · 2025-12-02
**Rebuttal Summary and Clarification (Part 1/2)**

Dear Area Chairs and Reviewers,

We sincerely thank the reviewers for the time and effort they devoted to the reviews and discussions. We are deeply regretful regarding the recent reviewer information leakage incident that affected the review process.

Below, we first provide a concise summary of the primary concerns shared across reviewers, along with the new experiments we conducted to further solidify our claims:

- **Generalization to Continuous Rewards (Reviewers 7NcB W4, o61x W1, wSTN W1):** Beyond binary-reward math tasks, we extended ExGRPO to RL settings with continuous preference-model rewards on the WildChat dataset. Results show that ExGRPO outperforms the GRPO baseline.

- **Efficiency Profiling (Reviewers 7NcB W1/Q1, o61x W1, B8SA Q2):** We provided detailed memory and time profiling on Qwen2.5-Math-7B/1.5B. The method introduces only moderate overhead (+2~19% training time) while providing substantial performance gains.

- **Design Heuristics (Reviewers 7NcB W2-3/Q2-4, o61x Q1, B8SA W1/Q1):** We conducted sensitivity analyses on Gaussian sampling and trajectory selection strategies, confirming that our design choices are robust and principled.

All remaining reviewer-specific questions were addressed thoroughly in the point-by-point rebuttal. For clarity, we also provide a structured per-reviewer summary below to help the new AC quickly navigate the key concerns and our corresponding responses.

Sincerely,

The Authors

---

> ### Author Response · Authors · 2025-12-02
> **Rebuttal Summary and Clarification (Part 2/2)**
>
> Dear Area Chairs and Reviewers,
>
> To support the newly assigned AC in understanding the rebuttal status, we summarize below the detailed issues raised by each reviewer and the corresponding evidence added during the rebuttal phase. A brief timeline of reviewers/authors engagement is also included for clarity.
>
> ---
>
> ### **Reviewer 7NcB**
>
> **Rating: 6**
>
> > **W1 (Q1).** Computational Overhead
> >
>
> **Our response & actions include:** clarifying that entropy is computed on a sampled subset only; profiling shows manageable cost (+1.77GB RAM, ~19% time for 7B; negligible for 1.5B).
>
> > **W2 (Q2).** Sensitivity of Medium-Difficulty Bucket
> >
>
> **Our response & actions include:** conducting a Gaussian-width sensitivity analysis; the default setting yields the best trade-off between math reasoning and general performance.
>
> > **W3 (Q3).** Replay Strategy
> >
>
> **Our response & actions include:** adding ablations confirming that *Lowest-Entropy* selection outperforms high-entropy and random sampling.
>
> > **W4.** Generalization to Continuous/Open-Domain Rewards
> >
>
> **Our response & actions include:** extending ExGRPO to continuous preference rewards (WildChat) using reward-variance bucketing; gains verified on ArenaHard and IFEval.
>
> > **Q4.** Retired Set Reintroduction
> >
>
> **Our response & actions include:** showing that reintroducing mastered queries slows convergence.
>
> ---
>
> ### **Reviewer o61x**
>
> **Rating: 6**
>
> > **W1.** Generalization to Continuous/Open-Domain Rewards. & Computational Overhead.
> >
>
> **Our response & actions include:** demonstrating reward-variance bucketing for continuous tasks; profiling shows modest overhead (2%–19% depending on model size).
>
> > **W2.** Baseline Positioning
> >
>
> **Our response & actions include:** clarifying that ExGRPO focuses on replay dynamics and is largely orthogonal to recent alignment baselines.
>
> > **Q1.** Gaussian Modeling Choice
> >
>
> **Our response & actions include:** providing empirical sensitivity evidence supporting medium-focused probabilistic sampling over heuristic bucketing.
>
> ---
>
> ### **Reviewer B8SA**
>
> **Rating: 4 → 8**
>
> > **W1 (Q1).** Design Heuristics. Symmetry vs Hard-Focus.
> >
>
> **Our response & actions include:** presenting ablations in terms of sampling and trajectory selection strategies ;conducting ablations showing that a hard-biased strategy (μ=0.25) underperforms the symmetric medium-focus design (μ=0.5).
>
> > **Q2.** Entropy Computation Efficiency
> >
>
> **Our response & actions include:** clarifying that entropy is computed on a per-batch subset, avoiding full-buffer recomputation.
>
> > **Q3.** Teacher Model Dependence
> >
>
> **Our response & actions include:** explaining that the teacher model was used only for early diagnostics; ExGRPO’s full training pipeline is self-contained.
>
> ---
>
> ### **Reviewer wSTN**
>
> **Rating: 8 → 8**
>
> > **W1 (Q1).** Generalization to Continuous/Open-Domain Rewards
> >
>
> **Our response & actions include:** validating ExGRPO on open-ended dialogue (WildChat) using reward variance as a difficulty proxy.
>
> > **Q2.** “Incorrect but Valuable” Trajectories
> >
>
> **Our response & actions include:** arguing that identifying such failures reliably requires a strong judge; instead, ExGRPO leverages naturally recurring failures in replay without external supervision.
>
> ---
>
> ### **Timeline of Rebuttal Updates**
>
> Due to the substantial compute required for adapting the codebase and running new large-scale experiments (especially for continuous-reward generalization), our complete rebuttal package was submitted on **24 Nov (AoE)**.
>
> Despite the short remaining window, two reviewers provided timely feedback:
>
> - **Nov 26 (AoE)** — Reviewer B8SA engaged actively and raised the score (4 → 8) **before the reviewer information leakage event**.
> - **Nov 27 (AoE)** — Reviewer wSTN confirmed that all concerns were resolved.
>
> ---
>
> We remain committed to full academic integrity and sincerely appreciate your efforts in maintaining a fair and rigorous evaluation environment under challenging circumstances.
>
> Sincerely,
>
> Authors

---

### Meta-Review · Area_Chair_P6Mj · 2026-01-06

**Summary:**

The paper propose to leverage experience replay within the modern RLVR machinery. In particular, they explore the idea of prioritizng datapoints from the replay buffer with respect to a heuristic defined on the accuracy and the entropy of the datapoints. The results show consistent gains, a deep analysis and a well-rounded comparison to numerous baselines.

Originally, reviewers had concerns about the computational cost of calculating entropy, but these were resolved as entropy is only recalculated for each minibatch. We could have questions whether this is the optimal strategy, but it is one that seemingly works. Additionally, reviewers asked whether the method was limited to verifiable domains, which the authors' rebuttal showed it wasn't.

Overall, this paper explores an important missing piece of current RLVR methods: replaying past experiences. It is not totally clear why the non-prioritized version of the algorithm does not bring strong gains. Questions may also linger as to why there isn't more results that show learning curves where sample efficiency is significantly increased. Indeed, this is one of the selling point of using experience replay in the deep RL era. Nonetheless, the paper provides a significant contribution that will benefit the community.

**Reviewer Concerns:**

Reviewer 7NcB and B8SA had concerns about the increased computational related to the entropy calculation. This was resolved in the rebuttal. Reviewer 7NcB, o61x and B8SA had concerns about the Gaussian heuristics used for selecting good datapoints. For most of the reviewers this would have likely been resolved. Finally, Reviewer wSTN had concerns whether this would generalize to non-verifiable domains. This is a fair concern but likely outside the scope of the paper. The authors still provided an initial experiment where they showed positive gains.

**Reviewer Scores:**

Reviewer B8SA would have increased their score and it is likely that Reviewer o61x would have as well.

---

### Decision · Program_Chairs · 2026-01-26

Accept (Poster)